# Characterization of Beeswax and Rice Bran Wax Oleogels Based on Different Types of Vegetable Oils and Their Impact on Wheat Flour Dough Technological Behavior during Bun Making

**DOI:** 10.3390/gels10030194

**Published:** 2024-03-12

**Authors:** Sorina Ropciuc, Florina Dranca, Mircea Adrian Oroian, Ana Leahu, Ancuţa Elena Prisacaru, Mariana Spinei, Georgiana Gabriela Codină

**Affiliations:** Faculty of Food Engineering, Stefan cel Mare University of Suceava, 720229 Suceava, Romania; sorina.ropciuc@fia.usv.ro (S.R.); florina.dranca@usm.ro (F.D.); m.oroian@fia.usv.ro (M.A.O.); analeahu@fia.usv.ro (A.L.); ancuta.prisacaru@fia.usv.ro (A.E.P.);

**Keywords:** oleogelation, physicochemical, rheological properties, DSC, rheofermentometer, Mixolab

## Abstract

Five varieties of vegetable oil underwent oleogelation with two types of wax as follows: beeswax (BW) and rice bran wax (RW). The oleogels were analyzed for their physicochemical, thermal, and textural characteristics. The oleogels were used in the bun dough recipe at a percentage level of 5%, and the textural and rheological properties of the oleogel doughs were analyzed using dynamic and empirical rheology devices such as the Haake rheometer, the Rheofermentometer, and Mixolab. The thermal properties of beeswax oleogels showed a melting peak at a lower temperature for all the oils used compared with that of the oleogels containing rice bran wax. Texturally, for both waxes, as the percentage of wax increased, the firmness of the oleogels increased proportionally, which indicates better technological characteristics for the food industry. The effect of the addition of oleogels on the viscoelastic properties of the dough was measured as a function of temperature. All dough samples showed higher values for G′ (storage modulus) than those of G″ (loss modulus) in the temperature range of 20–90 °C, suggesting a solid, elastic-like behavior of all dough samples with the addition of oleogels. The influence of the beeswax and rice bran oleogels based on different types of vegetable oils on the thermo-mechanical properties of wheat flour dough indicated that the addition of oleogels in dough recipes generally led to higher dough stability and lower values for the dough development time and those related to the dough’s starch characteristics. Therefore, the addition of oleogels in dough recipes inhibits the starch gelatinization process and increases the shelf life of bakery products.

## 1. Introduction

Food technologies use solid or semi-solid vegetable oils (shortenings and margarines) in different recipes to obtain baked goods that are soft and have high oxidative stability, an extended shelf life, and good sensory properties. However, their use in bakery product recipes is not desirable from a nutritional point of view due to its content of trans and saturated fatty acids, which may increase the risk of atherosclerosis, blood cholesterol, and other pathologies such as obesity, cardiovascular disease, cancer, and diabetes [1]. Their replacement with other types of fats is difficult to carry out in the food industry due to their technological characteristics, especially their solid properties at room temperature [2]. Nowadays, food producers and scientists are trying to obtain alternative fats without saturated and trans fatty acids that may be used in bakery products [3]. For this purpose, it is necessary to modify the liquid structure of oils into a solid one without changing their nutritional profile. Oleogelation may be a new alternative technology for obtaining semi-solid vegetable oils, and it involves structuring liquid fats using gelling agents; this technology has the potential to provide low saturated fat products with the solid structure that is required in the food industry [1,2,3]. Different types of oils may be used for oleogel production with a high level of biological activity of the lipids and a high content of essential fatty acids. Olive oil, grape seed oil, hemp seed oil, and walnut oil present a high nutritional value [4], but these oils cannot provide good technological characteristics to foods in their native state. Regarding the chemical composition of the oils, it is mainly composed of triglycerides (97–99%), which are mainly responsible for its biological properties and sensory attributes. Olive oil (OO) has a high content of MUFAs (65–83%), especially oleic acid and polyunsaturated fatty acids (PUFAs), such as linoleic acid, stearic acid, methyl octadecenoate C18:1(trans-9) +methyl oleate C18:1(cis-9), methyl linolelaidate, and methyl docosadienoic C23:0 (cis-16). Hemp seed oil (HO) contains linoleic acid (18:2 n-6) and α-linolenic acid (18:3 n-3) as its major omega-6 and omega-3 polyunsaturated fatty acids (PUFAs), respectively. Linolenic and linoleic fatty acids are known as essential fatty acids because humans cannot produce them themselves and must obtain them from their diet. Grape seed oil (GO) has a high content of PUFAs in the range of 85–90%. Oleic acid and monounsaturated acids (MUFAs) are also highly found in grape seed oil, and saturated fatty acids (SFAs) are present in lower quantities. The main fatty acids in grape seeds have linoleic acid C18:2 cis (n-6)-74.7% and C18:1 cis (n-9)-14.3% oleic acid. Some fatty acid contents of sunflower oil (SO) are palmitic acid 5.0–7.6%, stearic acid 2.7–6.5%, oleic acid 14.0–39.4%, linoleic acid 48.3–74.0%, and linolenic acid- 0–0.3% [5]. The fatty acid composition of walnut oils is as follows: palmitic 6.02–6.4%, palmitoleic 0.11–0.12%, stearic 2.65–2.92%, oleic 15.82–17.12%, linoleic 63.52–62.69%, linolenic 9.66–10.77%, arachidic (C20:0) 0.10–0.11%, and eicosanoid (C20:1) 0.16–0.19% [6]. For this, we propose that different oleogelators may be used for oleogel production with these types of oils such as phospholipids, sorbitan esters, fatty alcohols, mono- and diacylglycerols, ethylcellulose, phytosterols, and plant waxes, e.g., [7]. From these oleogelators, the use of plant waxes is of high interest due to their gel strength, oil binding capacity, abundance, and low cost. Natural waxes derived from seed coatings and plant cuticles as well as those secreted by insects are a mixture of hydrophobic compounds [8]. Different factors such as natural wax type, oil type, and wax concentration level may affect the characteristics of oleogels. For beeswax, the critical concentration of oleogel formation for most vegetable oils is around 6.0% *w*/*w* [9], whereas for rice bran it is 5.0% *w*/*w* [10,11]. An increase in wax levels decreases the gelation time. At the same time, the fatty acid composition of oils affects oleogel properties. For example, an oleogelator’s gelling ability may increase if the vegetable oil used for oleogel production presents a higher amount of saturated fatty acids in its composition [9,12]. According to our previous study, oleogels formulated with olive oil, grape seed oil, hemp seed oil, walnut oil, and sunflower oil using white beeswax (3%, 7%, and 11%) show high potential for saturated fat reduction in fat-based food products [3]. Oleogels have been developed in recent years, and the influence of replacing shortenings with them on the rheological and textural characteristics in various dough matrices, such as those of cookies, sponge cakes, shortcrust biscuits, sweet bread, or croissants, has been studied. According to different studies, the oleogels of candelilla wax with canola oil [7,10] and with rice bran oil reduced dough viscoelasticity [13,14,15]. Also, the same effect on dough viscoelasticity was reported for oleogels of sunflower oil with candelilla wax and beeswax wax; however, no negative effect on the final quality of bakery products was reported [8,16]. The hardness values for the dough samples increased when different oleogel types such as sunflower oil with carnauba and candelilla wax and canola oil with candelilla wax were incorporated in wheat flour [17]. The viscosity of short doughs decreased when the oleogels of sunflower oil and natural waxes (rice bran wax, beeswax, and candelilla wax) were added to their recipes [16,18]. To our knowledge, there are a few studies on the effects of oleogels on the textural and rheological characteristics of yeast-leavened food systems such as that of breads [2]. Moreover, dough’s rheological behavior during the bakery making process is a very complex one that implies different analyses during mixing, fermentation, and baking stages. Previous studies undertaken on the effect of the addition of oleogels into wheat flour on dough’s rheological properties are limited since they are focused only on some stages of bakery products process. Our study is a complete one carrying out a full analysis of dough’s rheological properties during the mixing, pasting, and fermentation process, which may anticipate the effect of oleogels on bakery’s technological process. Also, the aim of this study was to evaluate the effect of replacing shortenings with many types of oleogels such as wax/olive oil, grape seed oil, hemp seed oil, and walnut oil on the textural and rheological characteristics of yeast-leavened dough. These oleogels are diverse ones that may differently affect dough’s rheological characteristics. At the same time, the analysis of the oleogel properties that has been carried out during this study may be useful for understanding their behavior during the bakery technological process. This research is a complex and describes the preparation of oleogels from five different assortments of oil structured with beeswax and rice bran wax. The oleogels were described by analyzing the rheological and textural properties. Furthermore, the oleogels were incorporated according to a percentage level of 5% in the dough recipes of bakery products, and the dough’s rheological and textural behavior in these products was investigated.

## 2. Results and Discussion

Liquid oil was successfully structured into solid-like oleogels. The images of the oleogels obtained are shown in Figure 1. The rheological properties of oleogels that were influenced by the frequency and temperature, the melting and freezing behavior of the oleogels, and the textural properties were investigated.

### 2.1. Rheological Characterization of Oleogels according to Frequency

The frequency-dependent rheological properties were analyzed in order to describe the properties of oleogels. Rheology is one of the most powerful techniques capable of classifying a gelled system based on phase transitions and also of distinguishing between a true gel and a weak gel [19]. The classification of gels by oscillatory rheological measurements can be carried out for strong gels (G″/G′ ≤ 0.1) and not-so-strong gels (0.1 < G″/G′ < 1), which present less of a gel character [20,21]. Oleogels are systems that have a gel structure, and the condition that they must fulfill is that of the modulus of elasticity being higher than the modulus of viscosity. By comparison, according to Figure 2A,B, beeswax (BW) oleogels had better elastic properties compared with those of rice bran wax (RW) oleogels. At low applied frequency, the elastic component (G′) is always higher than the viscous component (G″) for gels and reaches a plateau in the linear viscoelastic region (LVR) [21,22]. With the increasing shear, the deformation of the gel structure takes place, and linearity is only kept in the range of 0.1–1 Hz, corresponding to the LVR range; then, the structure breaks. The moduli of the RW gel had a very low frequency dependence (more or less a linear curve) probably due to the fact that that it has a lower oil binding capacity than that of rice bran wax [20,23]. Moreover, the oleogels had higher G′ values than G″ ones, which indicate a solid, elastic-like behavior for all the samples. This behavior was in agreement with the results reported by Patel et al. [22] that also obtained lower loss modulus (G″) values than those of the storage moduli (G′) for different types of oleogels. However, the behavior of oleogels varies according to the function of the oil source type [24]. In principle, the fatty acid composition of the oil type regulates the oleogel’s flow behaviors. As may be seen, oleogels from grape seed oil, olive oil, and hemp oil present higher values for G′ and G″. According to Zhang et al. [25], oils with more double bonds may be loosely packed and more fluid-like. Also, G″ is highly correlated with the polyunsaturated fatty acid amount, which is at a high level in grape seed or hemp seed oil.

The behavior of oleogels at the melting temperature and the transition from the solid phase to the liquid one were analyzed from temperatures that varied in the range of 20–100 °C. The melting curves were approximately similar, and it may be seen that oleogels have a transition temperature of 40–50 °C (SO-5BW and HO-5BW). Similar results were obtained by Tavernier et al. [20] for sunflower oil and hemp seed oil oleogels with beeswax. They correlate the chemical structure of beeswax, which has long-chain wax esters, with a fatty acid portion between C20 and C24 as well as a fatty alcohol portion between C24 and C28, thereby leading to wax crystal formation at lower temperatures than those observed for other oleogelators [26].

The oleogels obtained with the rice bran wax have a very close transition point from the solid phase to the liquid phase. No difference was observed between samples formulated with different oils. An effective gelator must balance its solubility and insolubility in a solvent to achieve proper gelator–gelator and gelator–solvent interactions. The temperature difference observed for the oleogels with sunflower oil and beeswax as well as for the oleogel of hemp seed oil and beeswax at a percentage of 5%, thereby leading to the explanation that these combinations produce oleogels of poor quality. Fundamentally, a lipid structurant must impart solidity to the structured phase as well as prevent the exudation of the structured oil phase from the material. An alternative lipid structurant should exhibit physical properties (hardness/plasticity at a given temperature, melting point, melting profile) that are similar to those of the fat material it is intended to replace.

### 2.2. Thermal Properties of Oleogels

For the analysis of the thermal properties of oleogels, changes occurring between 20 °C and 90 °C, 90 °C and −60 °C, as well as −60 °C and 100 °C were considered; the values for the DSC parameters of oleogel samples, oils, and waxes—first heating and cooling profiles are presented in Table 1. The thermograms are showed in Figure 3 for oleogels, according to the function of the oil type that was used, for the five different oils, and they are showed separately for beeswax and rice bran wax; the values for the thermal parameters (onset, midpoint, end set, enthalpy, and heat capacity) are presented in Appendix A.

As can be observed in Figure 3 and in Appendix A in the Appendix A, the first heating profile (from 20 °C to 90 °C) of the oleogels and waxes showed similar thermal behaviors and contained one melting peak. As this peak was not found in the thermograms of the oil samples, any changes during this heating profile were attributed to the oleogelator (beeswax or rice bran wax). The melting peak was recorded at a lower temperature for oleogels with beeswax in comparison with oleogels containing rice bran wax, respectively. This was due to the lower melting temperature of beeswax when compared with that of rice bran wax, which was in accordance with the results of previous studies [11,27]. For all the oleogel samples, the melting peak was observed at a lower temperature than that of the melting peak of the wax used, and the melting enthalpy was lower when the proportion of wax in oleogels was 5%. The lower peak temperature and enthalpy at a lower oleogelator concentration were due to the dilution effect of the solvent, namely the oil used, which was also reported for oleogels with corn oil and rice bran oil [27,28].

The first cooling profile comprised two exothermic peaks, one corresponding to the phase transition, which was recorded for all oleogel samples and waxes, and one attributed to crystallization, which was only characteristic of oleogels containing olive oil and of the olive oil sample. Both BW and the oleogels with BW had a lower phase transition enthalpy in comparison with that of the RW oleogels and rice bran wax, indicating the lower crystallinity of this oleogelator, with these data being in agreement with previous studies [29]. The second exothermic peak that had a higher enthalpy was linked to the phase transition of the low melting of the highly unsaturated oil fraction of olive oil (oleic, linoleic, and palmitic acids) [30].

During the second heating and cooling profile, several endothermic and exothermic peaks were observed for the thermograms of the oleogel samples. Firstly, all the oleogels had a melting peak in the range between −40 and 0 °C, which was consistent with the endothermic peak that was recorded for the oil samples; this endothermic peak was also reported by other authors that studied the thermal behavior of vegetable oils [31,32] and was mostly related to the high content of low-melting triacylglycerols and the high proportion of mono- and polyunsaturated fatty acids that are the main components of the vegetable oils used to prepare oleogels [33]. The second melting peak was similar to the peak recorded in the first heating profile; however, the melting enthalpy was lower when compared with the first heating, thereby suggesting that after the first heating and cooling profile, the structure of the oleogel changed. In the second cooling phase, the same endothermic peak corresponding to the phase transition was found in all oleogel samples, and the enthalpy and heat capacity values were overall higher than those determined for the first cooling profile

### 2.3. Thermogravimetric Analysis of Oleogels

The thermal degradation of oleogels based on the oil type (grape seed, hemp, olive, sunflower, and walnut oil) and also the wax type (beeswax and rice bran wax) that was used for the formulation of them was studied by thermogravimetric analysis (TGA). Appendix A in the Appendix A presents the TGA curves, in the form of weight loss versus sample temperature, for the analyzed oleogels. Table 2 shows the values for the onset temperature (T_onset_) and the endset temperature (T_endset_) of the induced mass loss, the maximum weight loss rate (T_max_), the mass loss at the end of each decomposition (ΔW), and the residue that remains at 600 °C, which were calculated from thermograms using the method described by Sánchez et al. [34]. Excluding the initial mass loss at 150–200 °C, which is around 5%, as a result of water evaporation, another essential range is that oleogel decomposition took place in two well-defined stages. Therefore, the first stage (also known as the main stage) occurred between 295 and 480 °C due to the decomposition of the part of the oleogel molecules that depends on the percentage of saturated and unsaturated fatty acids present in the mass of total fat [35]. Moreover, this stage is also associated with the thermal volatilization of the vegetable oil used for the formulation of the oleogels [36]. The sample containing olive oil and 5% beeswax displays a slightly lower thermal stability (lower T_onset_, 297.26 °C), and the maximum degradation rate performed at a lower temperature (lower T_endest_, 451.10 °C). The second stage (between 480 and 555 °C) may be attributed to the degradation of the long-chain molecules (e.g., long-chain fatty acids, triacylglycerols) present in wax, increasing the possibility of crystallization [37]. The sample formulated with walnut oil and rice bran wax had the highest residue remaining at 600 °C (26.13%) in comparison with the beeswax sample; this can be explained by the fact that rice bran wax has a higher melting point compared with that of beeswax (this fact is supported by DSC analysis). The thermogravimetric data also show that the type of vegetable oil has a significant influence on the residue remaining at 600 °C, with lower values for olive oil and sunflower oil. Therefore, it can be observed that the incorporation of 9% beeswax and rice bran wax in the oleogel formulation has led to an improvement in thermal stability.

### 2.4. Texture Profile Analysis of Oleogels

The textural properties of oleogels (Table 3) are as important as the thermal properties and define the phenomena of gel stability and oil retention in the dough structure as well as their binding capacity. If the oleogels are of high hardness at a temperature of 20–22 °C, the possibility of incorporation into the dough is hindered. There is a risk that the structure of the oleogel does not allow for the incorporation of the oleogel during mixing and that it remains in the dough as particles. Of course, this can be prevented by bringing the oleogels to a temperature that ensures their complete incorporation into the dough mass. For both waxes, as the percentage of wax increased, the firmness of the oleogels increased proportionally (*p* < 0.001). Different values for the textural parameters were observed between the oil varieties used for oil freezing. For example, sunflower oil (SO), walnut oil (WO), and hemp seed oil with rice bran wax (RW) lead to oleogels with lower hardness (24–26 g) and higher stickiness (9.9–8.8%) compared with the samples obtained with grapeseed oil (GO) and olive oil (OL). The adhesiveness of the oleogel samples was close in value, and oleogels with higher stickiness were obtained in walnut oil and beeswax oleogels (WO-5BW) and also in sunflower oil and rice bran wax (RW) oleogels. The addition of 9% wax resulted in oleogels with higher hardness and lower stickiness. For most oleogels, it can be seen that the two types of wax form similar oleogels with different characteristics.

### 2.5. Rheological Characterization of Dough during Heating

The effects of the addition of oleogels at a 5% addition level in wheat flour on the viscoelastic properties of dough were measured as a function of temperature. All dough samples showed higher values for G′ (storage modulus) than those of G″ (loss modulus) in the temperature range of 20–90 °C (Figure 4). In the initial phase, the values for G′ and the G″ modulus decreased as the temperature increased, probably due to the protein denaturation that cannot retain water from the dough system. Gradually, the water release begins to be absorbed by starch granules that gelatinize, increasing the G′ and G″ values. As may be seen, the G′ values for the dough samples with the addition of oleogels with 5% beeswax are higher than those obtained for the dough samples with oleogels in which the percentage of wax was 9%. Jung et al. [11] formulated mixtures between rice bran oil and candelilla wax oleogels and introduced them into sweet breads, determining the extensibility of the dough accordingly. In the case of extensibility, the dough samples became more extensible with increasing oleogel levels in the dough recipe. In our research, the oleogel was used at a percentage of 5% in the leavened dough, which makes the results differ because of the type of oil used to formulate the oleogel, the wax used for oleogelation, and the percentage of the added wax in the wheat flour. In Figure 4A,C, the doughs with oleogel from sunflower oil (DSO-5BW) showed the highest values for the modulus of elasticity in the case of the dough samples with the oleogel with the beeswax addition. Comparatively, doughs with the oleogel from walnut oil (DWO-9RW) show high elastic behavior in samples formulated with 9% rice bran wax. The gelatinization temperature (60–80 °C) of the starch was influenced by the addition of the oleogel in the dough and not by the percentage of the added wax. The doughs obtained with rice bran wax (RW-9) showed the lowest values for the modules during heating. The values for the G′ G″ modules decreased with the addition of the oleogel with a wax content of 9%. This results is not an desirable one for fermented doughs because a tenacious dough with low elasticity cannot develop a product with a high specific volume and a crumb with fine porosity [38,39].

### 2.6. Rheological Properties of Dough during Fermentation

The rheological properties of wheat flour dough with an oleogel addition at a 5% addition level in wheat flour during fermentation are presented in Table 4. The dough samples with different oil and wax percentages of 5 and 9% showed significant variations in the maximum height of gas production (H’m). The H’m values for the samples with RW were between 42.20 mm and 47.11 mm compared with the dough samples to which oleogels based on beeswax (BW) were added, whereas the maximum height reached the value of 39.6 mm. Regarding the total volume of CO_2_ produced during fermentation (VT) and the volume of gas retained in the dough at the end of the test (VR), the highest values were obtained for the dough sample with oleogel obtained from olive oil and rice bran wax at a 9% addition (DOL-9RW), while the lowest ones were obtained for the dough samples with the oleogel obtained from sunflower seed oil and a 5% beeswax addition (DSO-5BW) and the oleogel with hemp seed oil and beeswax rice bran (DHO-5RW). The VT and VR values increased in the dough samples that contained oleogels with olive oil and 9% rice bran wax (RW) and beeswax (BW). This behavior is related to the viscous and elastic responses of the dough samples with the addition of oleogels, which may lead to a lower or a higher gas retention capacity. During dough fermentation, gas expansion increases the surface area and adsorption of lipid crystals. Crystalline lipids in oleogels with higher melting points that remain solid at the end of fermentation are the most effective at trapping gases. However, when too many lipids in solid form are present in the dough, its ability to grow during fermentation is inhibited [40]. Furthermore, the addition of oleogels containing solid lipids in an appropriate amount can strengthen the dough and thereby improve gas retention. The retention coefficient showed higher values for the dough samples obtained with oleogels from grape seed oil, walnut oil, and high wax BW/RW oleogels from olive oil and sunflower oil. It is well known that in bread making, lipids improve gas retention and thus increase its volume. According to Jung et al. [41], oleogels with solid-like properties are very effective in increasing the dough volume, which means it can improve the CR value and also the H’m and VT ones. Taking into account the fact that the CR is the ratio of the VR and VT values, it can be concluded that in some cases where the CR showed lower values compared with those of the control sample, the capacity of the dough to retain gases is lower than the amount of gases formed during the dough fermentation process [42].

### 2.7. Rheological Characterization of Dough from Wheat Flour and Different Types of Oleogels Using Mixolab

As Table 5 shows, different rheological data were obtained depending on the oleogel type used in the dough recipe. In the initial Mixolab mixing stage, stability (ST), dough development time (DDT), and C1 torque were determined. According to the data obtained, the DDT and ST are significantly different (*p* < 0.05) for the samples with the oleogel and shortening addition compared with those of the dough only obtained from wheat flour (control sample). Generally, compared with the control sample, the oleogel and shortening addition in wheat flour increased dough stability and decreased dough development time values. The sample with the shortening addition in wheat flour presented a similar behavior to that of the dough samples, which have oleogels incorporated in their recipe. Non-significant differences (*p* < 0.05) between the wheat flour dough with the 5% shortening addition (DWF-5SG) and the oleogel with olive oil and 5% beeswax (DOL-5BW) were obtained for the dough rheological properties of the mixing values. The increase in the stability value may be due to the lipid content from the oleogel and shortening samples, which may form lipoprotein complexes between the gluten, starch, and hydrophobic components, thereby leading to a more compact and stable dough [43]. Also, dough with the solid oleogel and shortening addition contain expandable and thin gluten films that favor air incorporation and dough stabilization during mixing [44]. The decrease in the DDT values with the oleogel addition is in agreement with those reported by Jung et al. [41] who concluded that a high amount of an oleogel addition to wheat flour dough leads to less resistance against mixing. The C1 torque value was 1.1 ± 0.04 Nm and represented the optimal dough consistency (C1 = 1.1 Nm) during mixing. Generally, when compared with the control sample, the Mixolab torques C2, C3, C4, and C5 decreased in the dough samples with the oleogel and shortening, and these data were in agreement with those reported by other authors [41,45]. According to them, vegetable oil dispersed all through the dough mixing in the form of oleogels or shortening, thereby leading to a soft dough that is less viscous and decreased the Mixolab torques during heating and cooling. These data are also in agreement with those reported by Oh et al. [46] who concluded that the addition of oleogels obtained using the organogelators of rice bran wax and beeswax reduced the viscosity of the dough samples, with this reduction being observed as higher for oleogels with rice bran wax compared with those obtained from beeswax. This different behavior may be due to the wax composition, which the in case of beeswax comprises 58% wax ester and more than 26% hydrocarbon, whereas in the case of rice bran wax it comprises 90% wax esters [15]. As the hydrogenated level of fat becomes higher, the dough becomes harder.

According to Baltsavias et al. [47], the hard fat may be spread to giant lumps during mixing, whereas the soft one is fragmented around the flour particles. Oils are less powerful in their oleogel system and may be dispersed during the mixing of all of them through the dough in the form of minuscule globules, thereby reducing dough consistency. The behavior of the dough samples during the gelatinization process changes significantly (*p* < 0.05) by adding different types of oleogels into wheat flour. It is generally accepted that in the presence of lipids when starch gelatinization takes place, the starch-amylose complexes have lower viscosity [45]. Molecular interaction takes place between amylose and the oleogels, thereby decreasing the leaching of amylose and reducing the swelling of starch, possibly leading to a reduction in the dough starch parameter values. This decreased all the Mixolab parameters related to the starch characteristics such as starch gelatinization (C3 and C32), cooking stability (C4), and starch gelling (C5 and C54). Through the oleogel or shortening addition in dough, some complexes between lipids and wheat flour compound protein and starch may be formed. This prevents the leaching of amylase from starch when it begins to gelatinize. Therefore, the swelling of the starch granules is delayed and the starch gelatinization process inhibited [48]. During the baking process, through the energy input, the dynamic equilibrium state changes, and the dough turns into bread through the starch gelatinization process and gluten denaturation network. Through the gelatinization process, starch changes its conformation, free glucan chains appear, and hydrophobic areas and hydrophilic surfaces change. Proteins occur as well as irreversible structural changes through thermal denaturation [44]. Under these conditions, there is a translocation of the lipids in the dough system. The lipids migrate from the gluten complex to the partially gelatinized starch, forming complexes with their main compound’s amylopectin and amylose, thus slowing down starch retrogradation after baking. This slowdown is reflected by an increase in bread shelf life indicated by the Mixolab decrease value of C54 due to the oleogel addition. The delay of the starch retrogradation process due to the addition of oleogels in the wheat flour dough, which prolongs the shelf life of bakery products, has also been reported by others [40,41,49].

### 2.8. Texture Profile Analysis of the Wheat Flour Dough with Different Types of Oleogel Additions

The textural parameters of the wheat flour dough with different types of oleogel additions at a 5% addition level in wheat flour are shown in Table 6. Significant differences (*p* < 0.05) in dough hardness and elasticity were obtained in samples with beeswax oleogels, especially in those with olive oil oleogels (DOL-9BW) and hemp seed oil oleogels (DHO-9BW) obtained with 9% wax. The results are in accordance with other works showing that the use of oleogels significantly affects the texture of the baked product, especially the hardness [48,49,50] and the elasticity of the final product, when oleogels are used as substitutes for traditional lipids. If the hardness value increases for doughs with an oleogel obtained with 9% wax, then the adhesiveness decreases significantly (*p* < 0.05). This behavior may be due to a higher level of SFAs in these types of oleogels that are fully saturated with hydrogen and present a more rigid and fixed structure being more solid-like and compact-packed in the dough system [25]. The doughs with the addition of 5% oleogel in its recipe were less sticky and were easier to process in a round format, specifically for buns. Cohesiveness was inversely proportional to adhesiveness. The dough samples with low adhesiveness (DGO-9BW; DHO-9BW) showed high cohesiveness values. Cohesion quantifies the internal strength of the food structure and provides information about the ability of a material to stick to itself [51]. Therefore, the oleogels analyzed in this study could be used in the fermented dough to improve the elasticity and hardness properties if they are formulated with a maximum of 5% added wax. Wax in a percentage of 9% used in the formulation of oleogels can lead to doughs with high hardness values.

### 2.9. Principal Component Analysis of Thermal and Rheological Properties of Oleogels

The graphical representation of the principal component analysis is shown in Figure 5. In this graph, a correspondence of the midpoint, enthalpy variables (DSC), and the transition temperature of the oleogels from the solid phase to the liquid phase (rheological properties) has been made. The graph of the main component’s groups for the oleogel samples according to the rheological properties and the transition point from the gel phase to the liquid phase has been made. According to the data obtained, the oleogels obtained with beeswax have a lower melting point compared with the oleogels obtained with rice bran wax. The inclusion of these oleogels in separate quadrants depends on the oleogelator used and the rheological properties of the oleogels. The oleogels with 5% rice bran wax correlate with the transition temperature from the gel to the liquid phase. The exception is the sunflower oleogel (SO) obtained with rice bran wax (SO-5RW) and the one with beeswax (SO-5BW), which do not have similar characteristics to the oleogels from the same group.

## 3. Conclusions

The oleogels used in the current study prepared on the base of the two wax assortments of BW and RW and five types of vegetable oils have the potential to be an alternative to unhealthy fats that are usually used in bakery industry. All the oleogels analyzed presented good quality characteristics for use in the food industry. The melting peak was recorded at a lower temperature for beeswax oleogels compared with that of the oleogels that contain rice bran wax. All the oleogel samples presented a solid, elastic-like behavior. The oleogel samples obtained with grape seed oil had higher viscoelasticity, followed by oleogels with hemp seed oil and oleogels with walnut oil. Regarding the dough rheological behavior, according to the data, a solid, elastic-like one was observed for all the samples across the whole range of frequencies. The addition of oleogels from olive oil and hemp seed oil in wheat flour dough led to high values for the G′ and G″ moduli, whereas the addition of oleogels based on walnut oil led to low values. Thermogravimetric analysis of oleogels described the decomposition of oleogels in two distinct stages as follows: the first stage corresponded to the decomposition of short-chain fatty and unsaturated acids at a temperature between 295 °C and 480 °C, and the second stage in the temperature range of 480–550 °C was associated with the thermal volatilization of the vegetable oil used to formulate oleogels. From the point of view of dough’s rheological properties during fermentation, doughs with rice bran wax presented better fermentation rheological values. The retention coefficient presented higher values for all dough samples, which indicated that the loaf volume of buns is adequate for all the samples with an addition of oleogels in their recipe. However, the use of oleogels based on walnut and hemp seed oil presented lower values for the CR parameter that indicated a lower capacity to retain gasses in the dough system when these types of oleogels were added. The results obtained with the Mixolab device indicated that the addition of oleogels significantly influenced the thermo-mechanical behavior of the wheat flour doughs. The use of oleogels in the dough system ensured a better stability during mixing, a decrease in the dough viscosity during heating and cooling, a delay of the starch gelatinization process, and better cooking stability and starch gelling. Our data indicated that the oleogels can improve the shelf life of the bakery products due to the fact that the difference between torques C5 and C4 decreased in the dough samples with oleogels in their recipe when compared with the control sample.

## 4. Materials and Methods

### 4.1. Materials

Beeswax and rice bran wax were supplied by Merck Romania. The vegetable oils of grape seed oil (GO), hemp seed oil (HO), olive oil (OL), sunflower oil (SO), and walnut oil (WO) were purchased from grocery stores. Wheat flour, type 650, was purchased from a local supplier. Yeast and salt were purchased from the local supermarket (Suceava, Romania).

### 4.2. Preparation of Oleogels

Oleogels were obtained using vegetable oils from 5 sources (olive, walnut, sunflower, grape seed, and hemp). Yellow beeswax and rice bran wax were used as oleogelation agents in percentages of 5% and 9%. These percentages have been chosen according to the literature where the main range of the level of the structure-forming agent in oleogels is from 0.5% *w*/*w* to 10.0% *w*/*w* [52,53]. For beeswax, the critical concentration of the oleogel formation for most vegetable oils is around 6.0% *w*/*w* [17], whereas for rice bran it is 5.0% *w*/*w* [10]. The oleogels were obtained according to the method described by Ropciuc et al. [3]. The oils and wax were heated to a temperature of 80 °C using a hotplate with an orbital stirrer until the wax completely dissolved. A total of 20 samples of oleogels were obtained for determinations. The mixture obtained was poured into tubes for solidification. The oleogels were stored under refrigerated conditions at 4 °C for five days until analysis. The samples were coded according to the type of oil used and the percentage of wax added to obtain the oleogel as follows: GO_5BW (oleogel from grape seed oil and 5% beeswax), HO_5BW (oleogel from hemp seed oil and 5% beeswax), OL_5BW (oleogel from olive oil and 5% beeswax), SO_5BW (oleogel from sunflower oil and 5% beeswax), and WO_5% (oleogel from walnut oil and 5% beeswax). The same form of coding was used for oleogels with a percentage of 9% beeswax. Rice bran wax samples were coded using the RW term (e.g., GO_5RW) [42,50,54].

### 4.3. Dough Preparation for Fundamental Rheological Analysis

Dough was obtained using flour type 650, salt, and water, into which 5% oleogel was incorporated according to the recipe described in Table 7. The hydration capacity of the flour was determined using the simulation test of the Mixolab device.

### 4.4. Dough Preparation for Rheological Properties Analysis during Fermentation

The dough samples for rheological property analysis during fermentation were obtained using type 650 flour, salt, yeast, water, and 5% oleogel.

### 4.5. Rheological Behavior of Oleogels and Dough

All rheological measurements were performed using a Haake Mars 40 Dynamic Rheometer (Karlsruhe, Germany) equipped with a Peltier system and circulating water bath (Huber, Germany) for temperature control. A glossy plate rotor with a diameter of 40 mm was used, with the distance between the plates being 1 mm. The linear viscoelastic region (LVR) of the oleogels was determined by logarithmic growth with the oscillation stress being from 0.1 to 100 Pa at a frequency of 1.0 Hz. The frequency sweep range was set from 0.1 to 10 Hz at a temperature of 25 °C. The viscoelastic properties of the oleogel samples were also evaluated under the influence of temperature. Modulus G′, “elastic” modulus G″, and “loss” were analyzed at temperatures between 20 and 90 °C, with a temperature increase of 2 °C/min according to the method previously described by Pușcaș et al. [4].

### 4.6. Differential Scanning Calorimetry

The influence of oleogel composition on thermal behavior was studied by differential scanning calorimetry (DSC). A DSC 25 calorimeter (TA Instruments, New Castle, DE, USA) calibrated with indium (28.71 J/g) at 156.6 °C was used for the analysis. An amount of approximately 4 mg for each sample was weighed and hermetically sealed in an aluminum pan that was inserted into the analysis instrument along with an empty pan used as a reference. The samples were first equilibrated at 20 °C, and then the temperature was varied in three steps as follows: from 20 °C to 90 °C, from 90 °C to −60 °C, and from −60 °C to 100 °C. The temperature was finally lowered to 20 °C. The heating rate was 10 °C/min and nitrogen was used as a purge gas at a flow rate of 20 mL/min.

### 4.7. Thermogravimetric Analysis of Oleogels

Thermogravimetric stability of the oleogels was evaluated using a thermal analysis system TGA 2 (Mettler Toledo, Columbus, OH, USA) by placing 5–10 mg of the sample in an aluminum oxide pan and subjecting it to heating from 30 to 600 °C at a heating rate of 10 °C/min in a nitrogen atmosphere (20 mL/min) according to the method previously described by Alvarez-Ramirez [55].

### 4.8. Texture Profile Analysis of Oleogels

To evaluate the textural properties of the oleogel samples with different oleogelation agents (beeswax and rice bran wax), a TVT-6700 texturometer (Perten Instruments, Hägersten, Sweden) was used with an aluminum cylinder of 25 mm diameter. The oleogel samples with a mass of 4 g were placed on a plastic plate with a diameter of 28 mm. The double compression method was used, with a compression percentage of 30%. The trigger force used was 10 g, the velocity was 5.0 mm/s, and the recovery period between compressions was 10 s.

### 4.9. Rheological Properties of Dough during Fermentation

The rheological properties of the dough during fermentation were determined using the Chopin Rheofermentometer F4 (Chopin, Villeneuve la Garenne, France) according to the Chopin protocol. The method used has been previously described by Atudorei et al. [56]. For this purpose, 315 g of dough was introduced into the curve of the Rheofermentometer, and it was pressed on top with a weight of 2 kg and subjected to fermentation at a temperature of 30 °C for 3 h. The parameters of dough formation, the maximum level of gas production (H’m), the volume of the gas retained in the dough at the end of the test (VR), the total volume of CO_2_ production (VT), and the retention coefficient (CR) were determined.

### 4.10. Rheological Characteristics of Dough Using Mixolab Device

The effect of different types of oleogel addition in wheat flour on the thermo-mechanical characteristics of dough was made using a Mixolab 2 device (Chopin SA., Villeneuve la Garenne, France) according to ICC Standard Method No. 173 [57]. Wheat flour with 5% of the different types of oleogel additions was mixed into the Mixolab bowl in which distilled water was added to obtain the optimum dough consistency. The protocol used for Mixolab analysis was the following: mixer temperature at 30 °C, kneading speed at 80 rpm, heating rate of 4 °C/min to 90 °C, and cooling rate of 4 °C/min to 50 °C.

### 4.11. Dough Texture Analysis

Dough texture analysis was evaluated using a TVT-6700 texture analyzer (Perten Instruments, Hägersten, Sweden) according to the method previously described by Jung et al. [41]. A dough with a mass of 50 g (according to the recipe presented in Table 7—dough components for rheological properties analysis during fermentation) was shaped into a sphere and placed on the platform, and a cylindrical compression platen probe of 35 mm diameter was used to perform a TPA test. The sample was compressed twice at a speed of 5 mm/s for 25 s followed by a rest time of 12 s between the two cycles and 10 mm/s retracting speed at a data rate of 200 pps. The textural parameters determined were hardness, stickiness, adhesiveness, cohesiveness, and springiness.

### 4.12. Statistical Analysis

Data reported in this study are mean values, and significant differences between samples were analyzed by one-way analysis of variance (ANOVA) by Tukey’s test (*p* < 0.05) using XLSTAT software, ver. 2023 (Addinsoft, New York, NY, USA). Pearson correlation (r) and principal component analysis (PCA) were also performed to determine relationships between different properties using XLSTAT software, ver. 2023.

## Figures and Tables

**Figure 1 gels-10-00194-f001:**
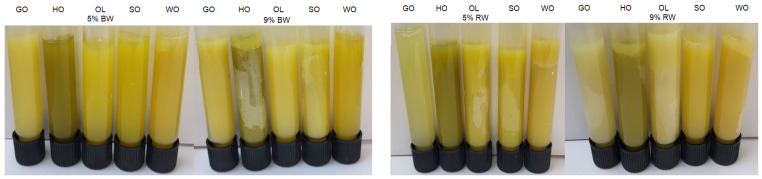
Visual appearance of vegetable oil oleogels with natural waxes.

**Figure 2 gels-10-00194-f002:**
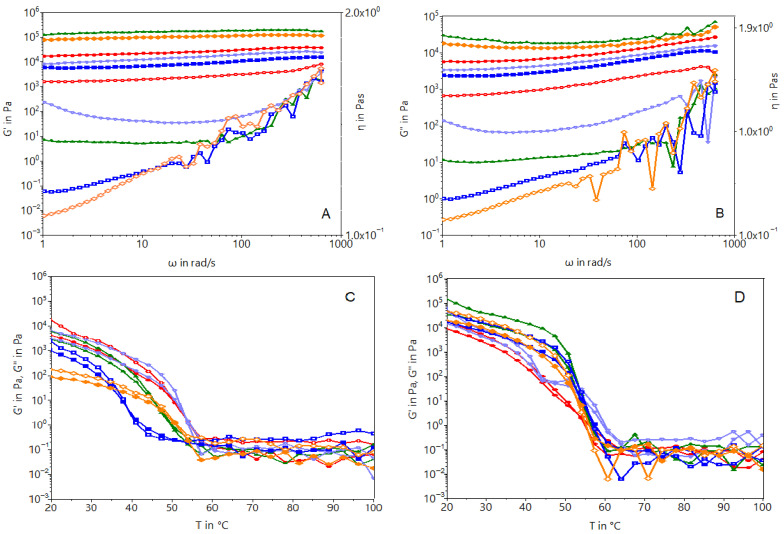
Effect of beeswax and rice bran concentrations on G′ and G″ with the angular frequency (**A**,**B**) and temperature (**C**,**D**). The red line represents the oleogel with grape seed oil (GO), the green line represents the oleogel with hemp seed oil (HO), the purple line represents the oleogel with olive oil (OL), the blue line represents the oleogel with sunflower oil (SO), and the orange line represents the oleogel with walnut oil (WO). Empty symbols represent the addition of 5% beeswax (BW) and solid symbols represent rice bran wax (RW). (**A**)—G′ angular frequency sweeps of oleogels with 5% wax; (**B**)—G″ angular frequency sweep of oleogels; (**C**)—oleogel modules G′ and G″ with 5% BW temperature sweep; (**D**)—oleogels modules G′ and G″ with 9% BW temperature sweep.

**Figure 3 gels-10-00194-f003:**
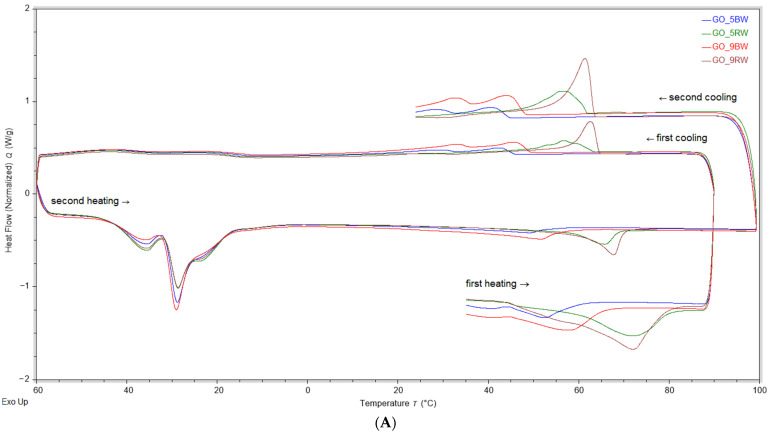
Thermograms of oleogels, oils, and waxes for heating (→) and cooling (←) profiles: (**A**) oleogels with grape oil, (**B**) oils, and (**C**) beeswax and rice bran wax.

**Figure 4 gels-10-00194-f004:**
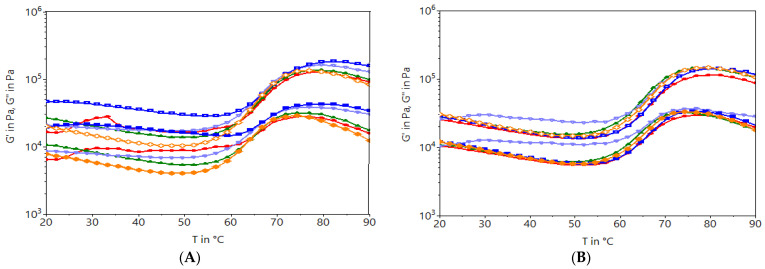
Graphical representation of the viscoelastic modules for dough samples with the addition of oleogels. (**A**,**B**)—describe the viscoelastic modules for doughs obtained with oleogels with 5% and 9% BW; (**C**,**D**)—represent the viscoelastic modules for doughs with the addition of 5% and 9% RW. The red line represents the dough with oleogel from grape seed oil (GO), the green line represents the dough with oleogel from hemp seed oil (HO), the purple line represents the dough with oleogel from olive oil (OL), the blue line represents the dough with oleogel from sunflower oil (SO), and the orange line represents the dough with oleogel from walnut oil (WO). The empty symbols represent the module G′ and the filled symbols represent G″.

**Figure 5 gels-10-00194-f005:**
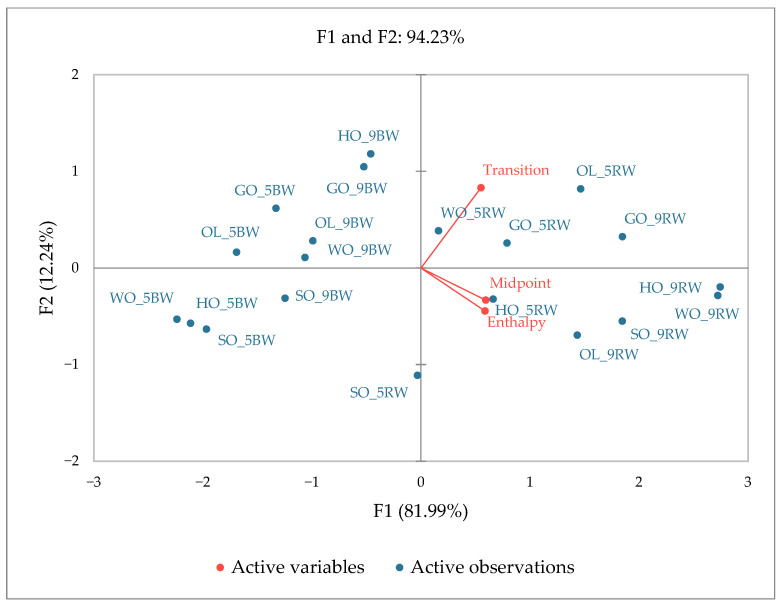
PCA graph for thermal and rheological properties of oleogels.

**Table 1 gels-10-00194-t001:** DSC parameters of oleogel samples, oils, and waxes—first heating and cooling profiles.

Sample	First Heating, 20 to 90 °C, Melting				First Cooling, 90 to −60 °C, Phase Transition					First Cooling, 90 to −60 °C, Phase Transition 2			
	Onset, °C	Midpoint, °C	End, °C	Enthalpy, J/g	Onset, °C	Midpoint, °C	End, °C	ΔCp, J/(g. °C)	Enthalpy, J/g	Onset, °C	Midpoint, °C	End, °C	Enthalpy, J/g
GO_5BW	43.85	52.47	64.41	2.29	48.82	45.81	43.56	0.0040	0.53	-	-	-	-
GO_9BW	44.71	58.39	68.95	4.52	52.61	49.28	46.59	0.0070	0.93	-	-	-	-
HO_5BW	44.79	53.82	64.10	1.57	49.37	46.10	42.83	0.0060	0.42	-	-	-	-
HO_9BW	42.87	52.98	64.85	4.44	52.82	48.96	46.93	0.0110	1.37	-	-	-	-
OL_5BW	44.05	52.47	61.56	2.28	47.78	45.15	43.70	0.0010	0.61	−57.97	−49.72	−41.35	24.51
OL_9B	43.74	53.85	64.76	4.91	50.96	47.98	46.34	0.0030	1.20	−57.63	−48.89	−39.81	28.76
SO_5BW	44.39	58.72	68.86	2.84	49.71	46.63	43.38	0.0010	0.35	-	-	-	-
SO_9BW	43.42	55.69	67.86	5.17	53.27	49.70	47.45	0.0020	1.22	-	-	-	-
WO_5BW	43.97	52.84	64.53	2.41	49.16	45.49	43.27	0.0050	0.78	-	-	-	-
WO_9BW	43.07	54.27	65.39	5.47	52.29	49.41	47.56	0.0060	0.87	-	-	-	-
GO_5RW	59.91	72.09	81.05	5.48	67.14	63.36	59.55	0.0110	1.00	-	-	-	-
GO_9RW	58.25	72.31	82.02	7.78	70.45	64.76	63.21	0.0120	5.56	-	-	-	-
HO_5RW	58.81	71.53	80.61	5.92	70.11	65.32	62.26	0.0160	2.61	-	-	-	-
HO_9RW	54.19	72.56	81.95	14.09	75.17	68.75	67.20	0.0270	6.36	-	-	-	-
OL_5RW	45.12	70.57	84.19	9.67	68.00	63.74	61.39	0.0080	1.71	−58.51	−50.41	−41.65	24.38
OL_9RW	44.04	70.50	84.56	14.77	69.51	62.66	61.38	0.0350	6.97	−58.43	−50.20	−42.45	27.82
SO_5RW	43.59	70.13	84.06	9.43	68.01	62.92	59.86	0.0020	1.38	-	-	-	-
SO_9RW	43.34	70.42	83.27	17.82	70.92	65.70	62.79	0.0150	4.34	-	-	-	-
WO_5RW	43.59	67.36	80.16	4.45	57.56	54.65	52.46	0.0001	0.53	-	-	-	-
WO_9RW	42.99	71.01	82.81	21.27	73.97	66.72	65.07	0.0160	11.13	-	-	-	-
GO	-	-	-	-	−16.43	−14.68	−12.59	0.0010	0.19	−53.28	−36.01	−31.80	7.39
HO	-	-	-	-	-	-	-	-	-	-	-	-	-
OL	-	-	-	-	−11.09	−13.88	−15.42	0.0020	1.23	−40.68	−48.84	−58.54	29.97
SO	-	-	-	-	-	-	-	-	-	-	-	-	-
WO	-	-	-	-	-	-	-	-	-	-	-	-	-
BW	34.68	67.28	83.91	185.05	67.53	60.49	59.34	0.0230	38.77	-	-	-	-
RW	77.12	82.00	84.44	20.71	81.96	76.27	75.47	0.0030	55.12	61.12	57.69	47.29	20.44

GO_5BW, GO_9BW, GO_5RW, and GO_9RW—oleogel with grape seed oil and 5% and 9% beeswax and, respectively, rice bran wax; HO_5BW, HO_9BW, HO_5RW, and HO_9RW—oleogel with hemp seed oil and 5% and 9% beeswax and, respectively, rice bran wax; OL-5BW, OL-9BW, OL-5RW, and OL_9RW—oleogel with olive oil and 5% and 9% beeswax and, respectively rice bran wax; SO_5BW, SO_9BW, SO_5RW, SO_9RW—oleogel with sunflower oil and 5% and 9% beeswax and, respectively, rice bran wax; WO_5BW, WO_9BW, WO_5RW, and WO_9RW—oleogel with walnut oil and 5% and 9% beeswax and, respectively, rice bran wax.

**Table 2 gels-10-00194-t002:** TGA parameters of oleogel samples.

Sample	T_onset_ (°C)	T_endset_ (°C)	T_final_ (°C)	ΔW (%)	Residue at 600 °C (%)
GO-5BW	323.11	473.48	484.76	83.89	16.11
GO-9BW	344.37	456.29	489.10	91.72	8.28
HO-5BW	355.90	464.09	494.47	86.56	13.44
HO-9BW	351.20	462.46	492.26	83.79	16.21
OL-5BW	297.26	451.10	497.00	93.96	6.04
OL-9BW	325.63	453.64	484.99	95.91	4.09
SO-5BW	365.27	467.64	499.52	96.37	3.63
SO-9BW	352.34	470.41	506.16	94.79	5.21
WO-5BW	347.49	478.81	519.74	93.00	7.00
WO-9BW	379.07	472.20	503.63	91.53	8.47
GO-5RW	355.46	470.95	503.00	92.55	7.45
GO-9RW	377.46	465.07	493.52	97.06	2.94
HO-5RW	378.64	471.00	498.89	89.61	10.39
HO-9RW	381.36	473.39	500.16	89.06	10.94
OL-5RW	360.34	463.31	555.45	97.09	2.91
OL-9RW	349.41	461.57	502.37	97.38	2.62
SO-5RW	351.67	467.95	503.95	97.66	2.34
SO-9RW	354.38	467.06	499.21	97.80	2.20
WO-5RW	380.29	472.22	513.74	73.87	26.13
WO-9RW	379.84	476.15	504.58	93.05	6.95

GO-5BW, GO-9BW GO-5RW, and GO-9RW—oleogel with grape seed oil and 5% and 9% beeswax and, respectively, rice bran wax; HO-5BW, HO-9BW, HO-5RW, and HO-9RW—oleogel with hemp seed oil and 5% and 9% beeswax and, respectively, rice bran wax; OL-5BW, OL-9BW, OL-5RW, and OL-9RW—oleogel with olive oil and 5% and 9% beeswax and, respectively, rice bran wax; SO-5BW, SO-9BW, SO-5RW, and SO-9RW—oleogel with sunflower oil and 5% and 9% beeswax and, respectively, rice bran wax; WO-5BW, WO-9BW, WO-5RW, and WO-9RW—oleogel with walnut oil and 5% and 9% beeswax and, respectively, rice bran wax.

**Table 3 gels-10-00194-t003:** TPA profile analysis of oleogels.

Sample	Hardness (g)	Springiness (%)	Stickiness (%)	Adhesiveness (g.s)	Cohesiveness (%)
GO-5BW	30 ^de^	0.9 ^ab^	−8.6 ^b^	−2.3 ^a^	0.63 c
HO-5BW	26 ^fg^	0.9 ^ab^	−9.81 ^b^	−3.8 ^a^	0.56 ^c^
OL-5BW	32 ^cd^	1.0 ^ab^	−7.8 ^ab^	−3.6 ^a^	0.55 ^d^
SO-5BW	30 ^de^	0.8 ^b^	−6.9 ^ab^	−2.8 ^a^	0.47 ^ef^
WO-5BW	29 ^def^	1.2 ^a^	−9.8 ^b^	−3.5 ^a^	0.42 ^f^
GO-9BW	32 ^cd^	0.9 ^ab^	−7.9 ^ab^	−3.4 ^a^	0.88 ^a^
HO-9BW	39 ^a^	1.1 ^ab^	−8.6 ^b^	−2.4 ^a^	0.82 ^a^
OL-9BW	37 ^ab^	1.0 ^ab^	−8.5 ^b^	−3.2 ^a^	0.78 ^a^
SO-9BW	31 ^cde^	1.0 ^ab^	−9.2 ^b^	−2.6 ^a^	0.84 ^a^
WO-9BW	30 ^cd^	0.8 ^b^	−9.7 ^b^	−2.7 ^a^	0.77 ^ab^
GO-5RW	36 ^ab^	1.2 ^a^	−8.5 ^b^	−3.1 ^a^	0.33 ^g^
HO-5RW	26 ^fg^	1.0 ^ab^	−8.6 ^b^	−3.8 ^a^	0.52 ^de^
OL-5RW	28 ^ef^	1.0 ^ab^	−9.0 ^b^	−2.9 ^a^	0.37 ^g^
SO-5RW	24 ^gh^	0.8 ^b^	−9.8 ^b^	−3.4 ^a^	0.41 ^f^
WO-5RW	26 ^fg^	0.8 ^b^	−8.5 ^b^	−3.6 ^a^	0.24 ^h^
GO-9RW	36 ^ab^	0.9 ^ab^	−8.6 ^b^	−3.1 ^a^	0.67 ^ab^
HO-9RW	32 ^cd^	0.9 ^ab^	−9.9 ^b^	−2.8 ^a^	0.42 ^f^
OL-9RW	34 ^bc^	1.1 ^ab^	−8.5 ^b^	−2.3 ^a^	0.36 ^g^
SO-9RW	36 ^ab^	1.2 ^a^	−9.1 ^b^	−2.4 ^a^	0.22 ^h^
WO-9RW	30 ^de^	1.0 ^ab^	−9.1 ^b^	−2.9 ^a^	0.21 ^h^
F-value	57.62 ***	4.82 ***	2.88 ***	0.58 ***	12.24 ***

GO_5BW, GO_9BW GO_5RW, and GO_9RW—oleogel with grape seed oil and 5% and 9% beeswax and, respectively, rice bran wax; HO_5BW, HO_9BW, HO_5RW, and HO_9RW—oleogel with hemp seed oil and 5% and 9% beeswax and, respectively, rice bran wax; OL_5BW, OL_9BW, OL_5RW, and OL_9RW—oleogel with olive oil and 5% and 9% beeswax and, respectively, rice bran wax; SO_5BW, SO_9BW, SO_5RW, and SO_9RW—oleogel with sunflower oil and 5% and 9% beeswax and, respectively, rice bran wax, WO_5BW, WO_9BW, WO_5RW, and WO_9RW—oleogel with walnut oil and 5% and 9% beeswax and, respectively, rice bran wax. Means in the same column with different letters for each sample type are significantly different (*p* < 0.05), and the F-value represents Tukey’s HSD test,*** strong correlation.

**Table 4 gels-10-00194-t004:** Rheological properties during fermentation of the dough samples.

Sample	H’m (mm)	VT (mL)	VR (mL)	CR (VR/VT), %
DGO-5BW	36.65 ^gh^	1250 ^cd^	1065 ^g^	88.32 ^ef^
DHO-5BW	38.36 ^fgh^	1263 ^cd^	1096 ^b^	86.71 ^f^
DOL-5BW	38,54 ^ef^	1131 ^fg^	1025 ^l^	90.64 ^cd^
DSO-5BW	38.36 ^efg^	986 ^j^	941 ^q^	95.50 ^a^
DWO-5BW	34.40 ^i^	1275 ^b^	1030 ^j^	80.80 ^h^
DGO-9BW	37.84 ^efgh^	1117 ^g^	1043 ^i^	93.30 ^b^
DHO-9BW	34.49 ^i^	1133 ^fg^	1013 ^n^	89.40 ^de^
DOL-9BW	33.99 ^i^	1239 ^d^	1090 ^d^	88.05 ^ef^
DSO-9BW	37.48 ^h^	1239 ^d^	1094 ^c^	88.32 ^ef^
DWO-9BW	39.60 ^e^	1212 ^e^	1079 ^e^	89.10 ^de^
DGO-5RW	45.21 ^a^	1117 ^g^	1043 ^i^	93.30 ^b^
DHO-5RW	42.92 ^bcd^	905 ^k^	750 ^r^	82.93 ^g^
DOL-5RW	44.51 ^ab^	1046 ^i^	985 ^o^	94.22 ^ab^
DSO-5RW	42.20 ^d^	1236 ^d^	1074 ^f^	86.94 ^f^
DWO-5RW	45.42 ^a^	1086 ^h^	1016 ^m^	93.63 ^b^
DGO-9RW	42.44 ^cd^	1127 ^g^	1028 ^k^	91.22 ^c^
DHO-9RW	44.23 ^abc^	1151 ^f^	1078 ^e^	90.81 ^cd^
DOL-9RW	46.00 ^a^	1364 ^a^	1220 ^a^	88.02 ^ef^
DSO-9RW	42.60 ^cd^	1209 ^d^	1062 ^h^	87.84 ^ef^
DWO-9RW	42.62 ^cd^	1049 ^i^	982 ^p^	93.51 ^b^
F-value	47.114 ***	262.27 ***	23,358.08 ***	42.57 ***

DGO_5BW, DGO_9BW DGO_5RW, and DGO_9RW—dough samples with 5% addition of oleogel with grape seed oil and 5% and 9% beeswax and, respectively, rice bran wax; DHO_5BW, DHO_9BW, DHO_5RW, and DHO_9RW—dough samples with 5% addition of oleogel with hemp seed oil and 5% and 9% beeswax and, respectively, rice bran wax; DOL_5BW, DOL_9BW, DOL_5RW, and DOL_9RW—dough samples with 5% addition of oleogel with olive oil and 5% and 9% beeswax and, respectively, rice bran wax; DSO_5BW, DSO_9BW, DSO_5RW, and DSO_9RW—dough samples with 5% addition of oleogel with sunflower oil and 5% and 9% beeswax and, respectively, rice bran wax; DWO_5BW, DWO_9BW, DWO_5RW, and DWO_9RW—dough samples with 5% addition of oleogel with walnut oil and 5% and 9% beeswax and, respectively, rice bran wax. Means in the same column with different letters for each sample type are significantly different (*p* < 0.05), and F-value represents Tukey’s HSD test,*** strong correlation.

**Table 5 gels-10-00194-t005:** Mixolab parameter values for wheat flour with different types of wax oleogel additions.

Samples	ST (min)	DDT (min)	C1 (N∙m)	C2 (N∙m)	C3 (N∙m)	C4 (N∙m)	C5 (N∙m)	C12 (N∙m)	C32 (N∙m)	C54 (N∙m)
WF	9.26 ± 0.05 ^g^	1.73 ± 0.05 ^i^	1.08 ± 0.03 ^ab^	0.48 ± 0.01 ^g^	1.73 ± 0.02 ^g^	2.20 ± 0.33 ^m^	4.43 ± 0.01 ^o^	0.59 ± 0.30 ^a^	1.24 ± 0.15 ^f^	2.23 ± 0.04 ^j^
WF-5SG	9.68 ± 0.10 ^k^	1.57 ± 0.02 ^fgh^	1.10 ± 0.01 ^ab^	0.46 ± 0.01 ^efg^	1.59 ± 0.01 ^f^	1.58 ± 0.01 ^f^	2.84 ± 0.01 ^d^	0.62 ± 0.02 ^ab^	1.13 ± 0.02 ^f^	1.26 ± 0.01 ^de^
DGO-5BW	9.56 ± 0.05 ^ijk^	1.12 ± 0.01 ^b^	1.13 ± 0.15 ^b^	0.45 ± 0.01 ^defg^	1.16 ± 0.02 ^e^	1.11 ± 0.01 ^d^	2.91 ± 0.01 ^h^	0.68 ± 0.17 ^bc^	0.70 ± 0.13 ^e^	1.79 ± 0.01 ^f^
DHO-5BW	8.80 ± 0.10 ^f^	1.59 ± 0.02 ^gh^	1.07 ± 0.02 ^ab^	0.44 ± 0.05 ^cdefg^	1.05 ± 0.01 ^bcde^	1.59 ± 0.01 ^g^	2.87 ± 0.00 ^def^	0.64 ± 0.55 ^abc^	0.62 ± 0.54 ^bcde^	1.28 ± 0.01 ^e^
DOL-5BW	9.66 ± 0.05 ^jk^	1.58 ± 0.02 ^fgh^	1.10 ± 0.01 ^ab^	0.46 ± 0.00 ^efg^	1.01 ± 0.01 ^bc^	1.66 ± 0.12 ^g^	2.81 ± 0.01 ^c^	0.65 ± 0.02 ^abc^	0.55 ± 0.01 ^b^	1.14 ± 0.01 ^b^
DSO-5BW	9.66 ± 0.05 ^jk^	1.59 ± 0.02 ^gh^	1.09 ± 0.01 ^ab^	0.44 ± 0.02 ^bcdef^	1.10 ± 0.00 ^bcde^	1.06 ± 0.00 ^c^	2.98 ± 0.01 ^i^	0.65 ± 0.28 ^abc^	0.66 ± 0.23 ^bcde^	1.91 ± 0.01 ^i^
DWO-5BW	9.70 ± 0.10 ^k^	1.58 ± 0.02 ^fgh^	1.10 ± 0.01 ^ab^	0.45 ± 0.00 ^efg^	1.01 ± 0.00 ^abc^	1.66 ± 0.00 ^gh^	2.80 ± 0.01 ^c^	0.65 ± 0.00 ^abc^	0.55 ± 0.01 ^b^	1.14 ± 0.01 ^b^
DGO-9BW	9.60 ± 0.10 ^jk^	1.47 ± 0.02 ^de^	1.09 ± 0.01 ^ab^	0.42 ± 0.00 ^bcdef^	1.58 ± 0.00 ^f^	1.52 ± 0.01 ^e^	2.72 ± 0.02 ^b^	0.67 ± 0.00 ^abc^	1.15 ± 0.00 ^f^	1.20 ± 0.15 ^c^
DHO-9BW	8.36 ± 0.05 ^e^	1.47 ± 0.02 ^de^	1.08 ± 0.01 ^ab^	0.41 ± 0.01 ^bcd^	1.12 ± 0.01 ^cde^	1.71 ± 0.12 ^j^	2.90 ± 0.01 ^gh^	0.68 ± 0.01 ^bc^	0.71 ± 0.02 ^e^	1.19 ± 0.01 ^c^
DOL-9BW	9.76 ± 0.05 ^k^	0.88 ± 0.02 ^a^	1.08 ± 0.01 ^ab^	0.42 ± 0.01 ^bcdef^	2.22 ± 0.02 ^h^	2.16 ± 0.01 ^l^	2.66 ± 0.01 ^a^	0.66 ± 0.14 ^abc^	1.80 ± 0.37 ^g^	0.49 ± 0.01 ^a^
DSO-9BW	4.30 ± 0.10 ^c^	1.49 ± 0.02 ^ef^	1.10 ± 0.01 ^ab^	0.40 ± 0.01 ^bc^	1.10 ± 0.01 ^bcde^	1.05 ± 0.01 ^c^	2.88 ± 0.01 ^defg^	0.70 ± 0.01 ^c^	0.70 ± 0.10 ^e^	1.82 ± 0.01 ^fg^
DWO-9BW	7.13 ± 0.05 ^d^	1.37 ± 0.03 ^cd^	1.07 ± 0.04 ^ab^	0.46 ± 0.02 ^efg^	1.03 ± 0.00 ^bcd^	0.97 ± 0.02 ^b^	3.45 ± 0.02 ^k^	0.62 ± 0.60 ^ab^	0.57 ± 0.17 ^bc^	2.48 ± 0.25 ^l^
DGO-5RW	9.33 ± 0.05 ^hi^	0.95 ± 0.02 ^a^	1.05 ± 0.03 ^a^	0.39 ± 0.02 ^b^	1.04 ± 0.01 ^bcd^	0.96 ± 0.01 ^b^	3.42 ± 0.02 ^j^	0.66 ± 0.15 ^abc^	0.65 ± 0.18 ^bcde^	2.45 ± 0.19 ^l^
DHO-5RW	8.95 ± 0.07 ^f^	1.60 ± 0.00 ^e^	1.06 ± 0.01 ^a^	0.40 ± 0.00 ^bc^	1.06 ± 0.00 ^bcde^	1.58 ± 0.12 ^f^	2.87 ± 0.01 ^de^	0.66 ± 0.12 ^abc^	0.65 ± 0.00 ^bcde^	1.28 ± 0.01 ^e^
DOL-5RW	3.70 ± 0.10 ^b^	1.20 ± 0.10 ^b^	1.13 ± 0.01 ^b^	0.31 ± 0.01 ^a^	1.03 ± 0.00 ^bcd^	1.87 ± 0.01 ^k^	4.15 ± 0.01 ^m^	0.83 ± 0.02 ^d^	0.72 ± 0.08 ^e^	2.27 ± 0.01 ^k^
DSO-5RW	9.66 ± 0.05 ^jk^	1.50 ± 0.02 ^efg^	1.09 ± 0.03 ^ab^	0.44 ± 0.02 ^bcdef^	1.12 ± 0.02 ^cde^	0.98 ± 0.00 ^b^	2.85 ± 0.01 ^d^	0.65 ± 0.53 ^abc^	0.67 ± 0.13 ^cde^	1.86 ± 0.01 ^h^
DWO-5RW	9.40 ± 0.10 ^hij^	1.45 ± 0.02 ^cde^	1.07 ± 0.02 ^ab^	0.43 ± 0.00 ^bcdef^	1.13 ± 0.01 ^de^	1.70 ± 0.01 ^hi^	2.95 ± 0.01 ^i^	0.64 ± 0.20 ^abc^	0.70 ± 0.01 ^e^	1.24 ± 0.01 ^de^
DGO-9RW	10.93 ± 0.15 ^l^	5.72 ± 0.03 ^j^	1.08 ± 0.03 ^ab^	0.46 ± 0.01 ^efg^	0.90 ± 0.01 ^a^	0.86 ± 0.012 ^a^	4.29 ± 0.01 ^n^	0.62 ± 0.27 ^ab^	0.43 ± 0.10 ^a^	3.43 ± 0.01 ^m^
DHO-9RW	7.40 ± 0.10 ^d^	5.73 ± 0.02 ^j^	1.10 ± 0.00 ^ab^	0.47 ± 0.00 ^fg^	2.46 ± 0.02 ^i^	2.38 ± 0.33 ^n^	3.59 ± 0.01 ^l^	0.64 ± 0.01 ^abc^	1.99 ± 0.00 ^h^	1.20 ± 0.01 ^c^
DOL-9RW	9.83 ± 0.05 ^k^	1.35 ± 0.01 ^c^	1.05 ± 0.01 ^a^	0.43 ± 0.00 ^bcdef^	1.01 ± 0.00 ^ab^	0.97 ± 0.01 ^b^	2.80 ± 0.01 ^c^	0.63 ± 0.01 ^abc^	0.57 ± 0.01 ^bcd^	1.83 ± 0.01 ^gh^
DSO-9RW	9.66 ± 0.15 ^k^	1.50 ± 0.02 ^efg^	1.07 ± 0.01 ^ab^	0.44 ± 0.01 ^bcdef^	1.03 ± 0.01 ^bcd^	0.98 ± 0.01 ^b^	2.85 ± 0.00 ^d^	0.63 ± 0.01 ^abc^	0.58 ± 0.01 ^bcd^	1.87 ± 0.01 ^h^
DWO-9RW	2.66 ± 0.05 ^a^	0.93 ± 0.02 ^a^	1.08 ± 0.03 ^a^	0.42 ± 0.00 ^bcde^	1.10 ± 0.01 ^bcde^	1.67 ± 0.01 ^gh^	2.89 ± 0.01 ^efg^	0.64 ± 0.01 ^abc^	0.68 ± 0.01 ^de^	1.21 ± 0.01 ^cd^

Mixolab parameters: ST, stability; DDT, dough development time; C1–C5, maximum consistency during stages 1–5, respectively; C12, difference between torques C1 and C2; C32, difference between torques C3 and C2; C34, difference between torques C3 and C4; C54, difference between torques C5 and C4. Samples: WF— wheat flour; WF-5SG—wheat flour with 5% shortening addition; DGO-5BW, DGO-9BW DGO-5RW, and DGO-9RW—dough samples with 5% addition of oleogel with grape seed oil and 5% and 9% beeswax and, respectively, rice bran wax; DHO-5BW, DHO-9BW, DHO-5RW, and DHO-9RW—dough samples with 5% addition of oleogel with hemp seed oil and 5% and 9% beeswax and, respectively, rice bran wax; DOL-5BW, DOL-9BW, DOL-5RW, and DOL-9RW—dough samples with 5% addition of oleogel with olive oil and 5% and 9% beeswax and, respectively, rice bran wax; DSO-5BW, DSO-9BW, DSO-5RW, and DSO-9RW—dough samples with 5% addition of oleogel with sunflower oil and 5% and 9% beeswax and, respectively, rice bran wax; DWO-5BW, DWO-9BW, DWO-5RW, and DWO-9RW—dough samples with 5% addition of oleogel with walnut oil and 5% and 9% beeswax and, respectively, rice bran wax. Means in the same column with different letters for each sample type are significantly different (*p* < 0.05).

**Table 6 gels-10-00194-t006:** TPA profile analysis of dough with oleogels.

Sample	Hardness (g)	Stickiness (%)	Adhesiveness (g.s)	Cohesiveness (%)	Springiness (%)
DGO_5BW	772 ^j^	−127.4 ^b^	−1294 ^k^	38.4 ^e^	1 ^NS^
DHO_5BW	729 ^p^	−216 ^gh^	−1453 ^l^	27.6 ^i^	1 ^NS^
DOL_5BW	845 ^d^	−203 ^e^	−2145 ^p^	34.2 ^f^	1 ^NS^
DSO_5BW	757 ^m^	−128 ^b^	−1876 ^n^	33.7 ^fg^	1 ^NS^
DWO_5BW	763 ^l^	−212 ^fg^	−1022 ^j^	40.1 ^de^	1 ^NS^
DGO_9BW	886 ^c^	−124 ^ab^	−890 ^g^	54.1 ^a^	1 ^NS^
DHO_9BW	954 ^a^	−236 ^ij^	−789 ^d^	56.8 ^a^	1 ^NS^
DOL_9BW	947 ^b^	−236 ^j^	−875 ^f^	47.5 ^b^	1 ^NS^
DSO_9BW	821 ^e^	−169 ^c^	−987 ^i^	40.2 ^de^	1 ^NS^
DWO_9BW	802 ^l^	−209 ^ef^	−908 ^h^	40.8 ^de^	1 ^NS^
DGO_5RW	766 ^k^	−189 ^d^	−2765 ^r^	33.4 ^fg^	1 ^NS^
DHO_5RW	751 ^n^	−190 ^d^	−2431 ^q^	30.7 ^ghi^	1 ^NS^
DOL_5RW	804 ^g^	−207 ^ef^	−1899 ^o^	31.5 ^fgh^	1 ^NS^
DSO_5RW	741 ^o^	−221 ^h^	−1765 ^m^	29.9 ^hi^	1 ^NS^
DWO_5RW	75 2^n^	−216 ^gh^	−2908 ^s^	27.6 ^i^	1 ^NS^
DGO_9RW	780 ^l^	−119 ^a^	−679 ^b^	38.3 ^e^	1 ^NS^
DHO_9RW	759 ^lm^	−125 ^ab^	−709 ^c^	38.7^de^	1 ^NS^
DOL_9RW	814 ^f^	−208 ^ef^	−814 ^e^	40.3 ^de^	1 ^NS^
DSO_9RW	847 ^d^	−230 ^i^	−712 ^c^	44.3 ^bc^	1^NS^
DWO_9RW	794 ^h^	−218 ^gh^	−608 ^a^	41.9 ^cd^	1 ^NS^
F-value	12,313.03 ***	1301.50 ***	1622 × 10^3^ ***	190.65 ***	

DGO_5BW, DGO_9BW DGO_5RW, and DGO_9RW—dough samples with 5% addition of oleogel with grape seed oil and 5% and 9% beeswax and, respectively, rice bran wax; DHO_5BW, DHO_9BW, DHO_5RW, and DHO_9RW—dough samples with 5% addition of oleogel with hemp seed oil and 5% and 9% beeswax and, respectively, rice bran wax; DOL_5BW, DOL_9BW, DOL_5RW, and DOL_9RW—dough samples with 5% addition of oleogel with olive oil and 5% and 9% beeswax and, respectively, rice bran wax; DSO_5BW, DSO_9BW, DSO_5RW, and DSO_9RW—dough samples with 5% addition of oleogel with sunflower oil and 5% and 9% beeswax and, respectively, rice bran wax; DWO_5BW, DWO_9BW, DWO_5RW, and DWO_9RW—dough samples with 5% addition of oleogel with walnut oil and 5% and 9% beeswax and, respectively, rice bran wax. Means in the same column with different letters for each sample type are significantly different (*p* < 0.05), and the F-value represents Tukey’s HSD test, *** strong correlation. NS—insignificant.

**Table 7 gels-10-00194-t007:** Dough components for rheological analysis.

Dough Components for Fundamental Rheological Determination				Dough Components for Rheological Property Analysis during Fermentation				
Flour, g	Salt, g	Water, mL	Oleogel, g	Flour, g	Salt, g	Yeast, g	Water, mL	Oleogel, g
30	0.45	17.4	1.5	300	5	3	174	15

## Data Availability

The original contributions presented in the study are included in the article/Appendix A, further inquiries can be directed to the corresponding authors.

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
