# Peer review of "Characterization of Beeswax and Rice Bran Wax Oleogels Based on Different Types of Vegetable Oils and Their Impact on Wheat Flour Dough Technological Behavior during Bun Making"

_gels, 2024, doi:10.3390/gels10030194_

Round 1
Reviewer 1 Report (New Reviewer)
Comments and Suggestions for Authors
General comments
Interesting study, well-written manuscript, and can be valuable for food scientists and the food industry. The rheological, thermal and textural properties of oleogels are influenced by the fatty acid composition of the oils used for their production (Pinto, Martins, Pastrana, Pereira, & Cerqueira, 2021). Therefore, it would be valuable for the study to indicate the fatty acid composition or at least the ratio of SFA to UFA in the oils to support the discussion.
All abbreviations in the tables and on the figures should be explained under the tables/figures. Tables and figures should be self-explanatory. In addition, the same abbreviations for oleogels and doughs with oleogels cause confusion. This should be corrected to avoid misunderstandings.
Specific comments:
Lines 77-79: Please provide the references for this statement.
Figure 1 should be cited in the text.
Line 128: Please cite Tavernier et al. correctly
Lines 150-154. It should be clarified that only thermograms for oleogels produced with grape seed oil and BW/RW are given.
Line 199. Where is Appendix C presented? Table 1B and 1C are only listed in supplementary material
In the sections dealing with doughs, the percentage of oleogels addition in the dough should be clearly stated to avoid misunderstandings.
In lines 292 and 294, the words “lager” and “small” amount are not appropriate, please replace with “9%” and “5%”.
Table 3 and Table 4. What does *** mean in the F-values? It should be explained below the table.
Line 320: What is the full name of WF_5SG? This abbreviation appears for the first time.
Lines 329-331: Unclear sentence, please rephrase.
Lines 382-383: Please check the sentence, doughs with 9% oleogels or doughs with 4.16% oleogels formulated with 9% waxes (Table 7, Dough components for rheological properties analysis during fermentation)???
Line 488: wo changed to WO
Line 554: Table 1? You mean Table 7?
Pinto, T. C., Martins, A. J., Pastrana, L., Pereira, M. C., & Cerqueira, M. A. (2021). Oleogel-Based Systems for the Delivery of Bioactive Compounds in Foods. Gels, 7(3), 86. https://www.mdpi.com/2310-2861/7/3/86.
Author Response
28 February 2024
Dear Referee,
We would like to thank the referee for the close reading and for the proper suggestions. We hope that we provide all the answers to the reviewer’s comments.
Thank you very much for the recommendations to publish our paper entitled “Characterization of beeswax and brown rice wax oleogels based on different types of vegetable oils and their impact on wheat flour dough technological behavior during buns making”.
The present version of the paper has been revised according to the reviewer’s suggestions.
We uploaded the corrected version of the article for which we used the red color for the addition text.
Referee comments: Interesting study, well-written manuscript, and can be valuable for food scientists and the food industry. The rheological, thermal and textural properties of oleogels are influenced by the fatty acid composition of the oils used for their production (Pinto, Martins, Pastrana, Pereira, & Cerqueira, 2021). Therefore, it would be valuable for the study to indicate the fatty acid composition or at least the ratio of SFA to UFA in the oils to support the discussion.
Response: We would like to thank to the referee for his/her close reading of our manuscript data and his/her suggestions. We agree with the referee point of view and we added in the introduction part informations related to oleogels fatty acids composition. Also, we introduced in the discussion part some informations related between oleogels and dough behavior and the fatty acids content of the oleogels used. We also used as the reference the article of Pinto et al., 2021.
Referee comments: All abbreviations in the tables and on the figures should be explained under the tables/figures. Tables and figures should be self-explanatory. In addition, the same abbreviations for oleogels and doughs with oleogels cause confusion. This should be corrected to avoid misunderstandings.
Response: We would like to thank to the referee for his/her suggestions. We explained all abbreviations under the tables and we changed the dough abbreviations in order to avoid confusions.
Referee comments: Lines 77-79: Please provide the references for this statement.
Response: We provided.
Referee comments: Figure 1 should be cited in the text.
Response: We cited.
Referee comments: Line 128: Please cite Tavernier et al. correctly
Response: We cited now correctly.
Referee comments: Lines 150-154. It should be clarified that only thermograms for oleogels produced with grape seed oil and BW/RW are given
Response: We would like to thank to the referee for the close reading of our manuscript. Thermograms for HO, OL, SO, WO oils can be found in Supplementary materials S1 Figure 1. We clarified now in the manuscript.
Referee comments: Line 199. Where is Appendix C presented? Table 1B and 1C are only listed in supplementary material
Response: We want to tahbk to the referee for the close reading of our manuscript. We atatched now and appendix C which also exist.
Referee comments: In the sections dealing with doughs, the percentage of oleogels addition in the dough should be clearly stated to avoid misunderstandings.
Response: We completed now in the manuscript the addition level of oleogels in wheat flour dough (5%). It is also mentioned in abstract.
Referee comments: In lines 292 and 294, the words “lager” and “small” amount are not appropriate, please replace with “9%” and “5%”.
Response: We replaced.
Referee comments: Table 3 and Table 4. What does *** mean in the F-values? It should be explained below the table.
Response: We completed our manuscript with F explanation under the table.
Referee comments: Line 320: What is the full name of WF_5SG? This abbreviation appears for the first time.
Response: We completed now in the manuscript the full name of WF_5SG which we changed with DWF_5SG.
Referee comments: Lines 329-331: Unclear sentence, please rephrase.
Response: We rephrased it.
Referee comments: Lines 382-383: Please check the sentence, doughs with 9% oleogels or doughs with 4.16% oleogels formulated with 9% waxes (Table 7, Dough components for rheological properties analysis during fermentation)???
Response: We want to thank to the referee for the close reading of our manuscript. Indeed was a mistake. The addition of oleogel was of 5% in dough recipe (we corrected in the manuscript now). Also we corrected in the table 7.
Referee comments: Line 488: wo changed to WO
Response: We changed.
Referee comments: Line 554: Table 1? You mean Table 7?
Response: We want to thank to the referee for the close reading of our manuscript. Indeed was a mistake. Correct was Table 7. We corrected now in the manuscript.
Referee comments: Pinto, T. C., Martins, A. J., Pastrana, L., Pereira, M. C., & Cerqueira, M. A. (2021). Oleogel-Based Systems for the Delivery of Bioactive Compounds in Foods. Gels, 7(3), 86. https://www.mdpi.com/2310-2861/7/3/86.
Response: We completed the manuscript with this reference.
Sincerely,
Georgiana Codină et al.

Reviewer 2 Report (New Reviewer)
Comments and Suggestions for Authors
Characterization of beeswax and brown rice wax oleogels based 2 on different types of vegetable oils and their impact on wheat 3 flour dough technological behaviour during buns making
The author properly justified the comments of the reviewer.
Well, the manuscript written is good and the author please incorporate my suggestion to enhance the effectiveness of this manuscript for the esteemed journal.
Comment 1: Language is not up to the mark (Scientific approach required) in introduction please rewrite it
Comment 2: Use the significantly (p<0.05) and non-significantly (p<0.05) term in
the manuscript
Comment 3: Compare the results with the previous studies and there is no justification of result so must add
Comment 4: Add citation on methodology part
Comment 5: Use more scientific terms in whole manuscript
Comments on the Quality of English LanguageMinor editing of the English language required
Author Response
28 February 2024
Dear Referee,
We would like to thank the referee for the close reading and for the proper suggestions. We hope that we provide all the answers to the reviewer’s comments.
Thank you very much for the recommendations to publish our paper entitled “Characterization of beeswax and brown rice wax oleogels based on different types of vegetable oils and their impact on wheat flour dough technological behavior during buns making”.
The present version of the paper has been revised according to the reviewer’s suggestions.
We uploaded the corrected version of the article for which we used the red color for the addition text.
Referee comments: The author properly justified the comments of the reviewer.
Well, the manuscript written is good and the author please incorporate my suggestion to enhance the effectiveness of this manuscript for the esteemed journal.
Response: We would like to thank the referee for his/her appreciation, close reading of our manuscript data and his/her suggestions.
Referee comments: Comment 1: Language is not up to the mark (Scientific approach required) in introduction please rewrite it
Response: We would like to thank to the referee for his/her suggestions. We rewrited the paragraphs according to the referee suggestions.
Referee comments: Use the significantly (p<0.05) and non-significantly (p<0.05) term in the manuscript
Response: We revised in our manuscript according to the referee suggestions.
Referee comments: Compare the results with the previous studies and there is no justification of result so must add
Response: We completed our manuscript with more justifications about the results obtained according to the referee suggestions.
Referee comments: Add citation on methodology part
Response: We completed the methodology part with citations.
Referee comments: Use more scientific terms in whole manuscript
Response: We used more scientific terms in whole manuscript according to the referee suggestions.
Sincerely,
Georgiana Codină et al.

Reviewer 3 Report (New Reviewer)
Comments and Suggestions for Authors
This is a very interesting manuscript. However, there are some issues with the use of english language and the terminology of the methods
For the enlish language use please rephrase/correct the following : 44-47, 57, 70-71, 77-79, 92-94, 97, 99-102, 311-312, 440-441, 446-465.
line 59: brown rice?
line 75: cookies doughs
line 141-147: which temperature difference and how is that related to what reported by the cited reference?
line 240: tartrate wax?
paragraph 2.4: you are using the term tackiness, which is not explained or introduced somewhere in the manuscript
line 382: 9% oleogel?
llines 477-479: this is repetition
line 493: oleogel was incorporated in water?
line 109-111: In rheology, the only correct term for gels is "true gels" (which can be strong or less strong). The term "weak gel" is used only for special behaviours observed e.g. for xanthan.
Moreover, G'' is not the plastic modulus but the loss modulus. Frequency is denoted as ω and not f.
Please relate the results from the rheology experiments to those of the DSC.
For the frequency sweeps the GO, OL and SO oleogels differed significantly regarding the wax used. Can you comment on that?
What was the necessity in using two dough recipes?
Comments on the Quality of English LanguageAs stated earlier, there are some issues with the use of english language (see suggested corrections)
Author Response
28 February 2024
Dear Referee,
We would like to thank the referee for the close reading and for the proper suggestions. We hope that we provide all the answers to the reviewer’s comments.
Thank you very much for the recommendations to publish our paper entitled “Characterization of beeswax and brown rice wax oleogels based on different types of vegetable oils and their impact on wheat flour dough technological behavior during buns making”.
The present version of the paper has been revised according to the reviewer’s suggestions.
We uploaded the corrected version of the article for which we used the red color for the addition text.
Referee comments: This is a very interesting manuscript. However, there are some issues with the use of english language and the terminology of the methods
Response: We would like to thank the referee for his/her close reading of our manuscript. We improved our manuscript according to the referee suggestions.
Referee comments: For the English language use please rephrase/correct the following : 44-47, 57, 70-71, 77-79, 92-94, 97, 99-102, 311-312, 440-441, 446-465.
Response: We corrected (rephrased) the paragraphs mentioned by referee.
Referee comments: line 59: brown rice?
Response: We want to thank to the referee for pointing out this mistake. Correct was bran rice. We revised it in the mansucript.
Referee comments: line 75: cookies doughs
Response: We changed with the term short doughs. The cookies are the final product.
Referee comments: line 141-147: which temperature difference and how is that related to what reported by the cited reference?
Response: We have now completed in the manuscript more informations related to the temperature difference and how is related to the cited reference.
Referee comments: line 240: tartrate wax?
Response: We want to thank to the referee for the close reading of our manuscript. It was a mistake. The correct term was bran rice. We revised.
Referee comments: paragraph 2.4: you are using the term tackiness, which is not explained or introduced somewhere in the manuscript
Response: We want to thank to the referee for the close reading of our manuscript. It was a mistake. The correct term was stickiness. We revised.
Referee comments: line 382: 9% oleogel?
Response: Wax was used in 9% for oleogel production.
Referee comments: lines 477-479: this is repetition
Response: We revised.
Referee comments: Line 493: oleogel was incorporated in water?
Response: Lines 492-495 describe how to obtain the dough. The oleogel was added to the dough and incorporated with the ingredients for the fermented dough (flour, yeast, salt and water)
Referee comments: line 109-111: In rheology, the only correct term for gels is "true gels" (which can be strong or less strong). The term "weak gel" is used only for special behaviours observed e.g. for xanthan.
Response: We revised the terms in the manuscript according to the referee suggestions.
Referee comments: Moreover, G'' is not the plastic modulus but the loss modulus. Frequency is denoted as ω and not f.
Response: We made the graphs function of frequency (Hz) and not angular frequency (ω (rad/s))
Referee comments: Please relate the results from the rheology experiments to those of the DSC.
Response: We made another PCA graph in order to comply with the referee suggestions which reflect this correlations.
Referee comments: For the frequency sweeps the GO, OL and SO oleogels differed significantly regarding the wax used. Can you comment on that?
Response: We completed in the manuscript some explanations related to the rehology data and how these were affected by oleogels types.
Referee comments: What was the necessity in using two dough recipes?
Response: Fro dough samples we used only one recipe. The other recipe is for oleogel production.
Sincerely,
Georgiana Codină et al.

Round 2
Reviewer 2 Report (New Reviewer)
Comments and Suggestions for Authors
All changes are done by the author. Now manuscript was ready for further step
Author Response
We would like to thank to the referee for his/her acceptance and appreciations.
Reviewer 3 Report (New Reviewer)
Comments and Suggestions for Authors
The authors have made corrections to their manuscript but there are still some minor issues that need to be addressed
1. English language: rephrase e.g. lines 139-142, 338-340
2. rheology: lines 126-129: As I wrote in my previous comments there are true gels and weak gels. Weak gels are referring to a few special systems, which are not found in the present study. So you only have true gels that can be either strong or not so strong (less gel character). Moreover, viscous solids are not classified under the term "gel". Regarding frequency, since you are using a rheometer is angular frequency and is always denoted as ω.
Comments on the Quality of English Language
minor mistakes
Author Response
5 March 2024
Dear Referee,
We would like to thank the referee for the close reading and for the proper suggestions. We hope that we provide all the answers to the reviewer’s comments.
Thank you very much for the recommendations to publish our paper entitled “Characterization of beeswax and brown rice wax oleogels based on different types of vegetable oils and their impact on wheat flour dough technological behavior during buns making”.
The present version of the paper has been revised according to the reviewer’s suggestions.
We uploaded the corrected version of the article for which we used the green color for the addition text.
Referee comments: 1. English language: rephrase e.g. lines 139-142, 338-340
Response: We corrected (rephrased) the paragraphs mentioned by referee.
Referee comments: 2. rheology: lines 126-129: As I wrote in my previous comments there are true gels and weak gels. Weak gels are referring to a few special systems, which are not found in the present study. So you only have true gels that can be either strong or not so strong (less gel character). Moreover, viscous solids are not classified under the term "gel". Regarding frequency, since you are using a rheometer is angular frequency and is always denoted as ω.
Response: We want to thank to the referee for pointing out this mistake. We corrected in the mansucript according to the referee suggestions, and we also changed the graphs with angular frequency instead of frequency.
Sincerely,
Georgiana Codină et al.

This manuscript is a resubmission of an earlier submission. The following is a list of the peer review reports and author responses from that submission.
Round 1
Reviewer 1 Report
Comments and Suggestions for Authors
The manuscript “Characterization of beeswax and brown rice wax oleogels based 2 on different types of vegetable oils and their impact on wheat 3 flour dough technological behavior during buns making” is generally well-oriented, the results are well-investigated. The manuscript is highly recommended to be published after some revision and answering the following questions.
1- Abstract must be enriched via valuable results which pave the way for understanding the audiences.
2- The introduction is written with poor information and did not cover the importance of this topic. So, the authors are requested to include a detailed and comprehensive study of an application and report the novelty of this work in the introduction section.
3- Authors should perform the thermal gravimetric analysis (TGA) test to strength the quality of this paper.
4- XRD test is recommended to be performed to investigate the crystallinities of samples.
5- Conclusion is short and lacks the basic fundamentals of the results obtained. Please, the authors should re-write the conclusions again with more emphasis on the significant comparison and the improvements from the results obtained.
Comments on the Quality of English Language
-
Author Response
14 December 2023
Dear Referee,
We would like to thank the referee for the close reading and for the proper suggestions. We hope that we provide all the answers to the reviewer’s comments.
Thank you very much for the recommendations to publish our paper entitled “Characterization of beeswax and brown rice wax oleogels based on different types of vegetable oils and their impact on wheat flour dough technological behavior during buns making”.
The present version of the paper has been revised according to the reviewer’s suggestions.
We uploaded the corrected version of the article for which we used the red color for the addition text and strikethrough for the delete text.
Reviewer comments
The manuscript “Characterization of beeswax and brown rice wax oleogels based on different types of vegetable oils and their impact on wheat flour dough technological behavior during buns making” is generally well-oriented, the results are well-investigated. The manuscript is highly recommended to be published after some revision and answering the following questions.
Response: We want to thank to the referee for the close reading of our manuscript. We revised the manuscript according to the referee suggestions. Also, would like to thank the reviewer for all the comments and suggestions which have helped us to improve our paper.
Reviewer: Abstract must be enriched via valuable results which pave the way for understanding the audiences.
Response: We would like to thank to the referee for his/her suggestions. We enriched the abstract in order to be better understand by the audience.
Reviewer: The introduction is written with poor information and did not cover the importance of this topic. So, the authors are requested to include a detailed and comprehensive study of an application and report the novelty of this work in the introduction section.
Response: We would like to the referee for his/her remark. We agree with the referee point of view and we revised the introduction part.
Reviewer: Authors should perform the thermal gravimetric analysis (TGA) test to strength the quality of this paper.
Response: According to referee suggestions, we completed our manuscript with thermal gravimetric analysis (TGA) in order to strength the quality of this paper.
Reviewer: XRD test is recommended to be performed to investigate the crystallinities of samples.
Response: We want to thank to the referee for his/her suggestions. Unfortunately, we do not have the possibility to make this analysis to our university.
Reviewer: Conclusion is short and lacks the basic fundamentals of the results obtained. Please, the authors should re-write the conclusions again with more emphasis on the significant comparison and the improvements from the results obtained.
Response: We revised the conclusions part according to referee suggestions.
Sincerely,
Georgiana Codină et co.
Reviewer 2 Report
Comments and Suggestions for Authors
This paper describes the use of oleogels with beeswax and brown rice bran wax in a dough recipe for bakery products. The authors analyze the rheolegical and textural properties of the oleogels and dough. The paper is well-written and it deals with a very interesting field, which the use of oleogels in dough. I believe that it will be of interest to the readers of this journal. Having said that, there are some issues that need to be addressed in order to improve the quality and readability of the paper. These issues are the following:
1. In lines 92 - 97, the authors state that there are only a few works about the effects of oleogels on the textural and rheological characteristics of yeast-leavened food systems such as breads. However, the authors need to state what's the novelty of their work compared to these works.
2. Except for the references for oleogels in dough, the paper should include a paragraph about oleogels with waxes. Some of the latest works that can be used as a starting point are the following:
- Frolova, Y.; Sarkisyan, V.; Sobolev, R.; Makarenko, M.; Semin, M.; Kochetkova, A. The influence of edible oils’ composition on the properties of beeswax-based oleogels. Gels 2022, 8, 48.
- Dimakopoulou-Papazoglou, D.; Giannakaki, F.; Katsanidis, E. Structural and Physical Characteristics of Mixed-Component Oleogels: Natural Wax and Monoglyceride Interactions in Different Edible Oils. Gels 2023, 9, 627.
- Winkler-Moser, J.K.; Anderson, J.; Felker, F.C.; Hwang, H.-S. Physical properties of beeswax, sunflower wax, and candelilla wax mixtures and oleogels. J. Am. Oil Chem. Soc. 2019, 96, 1125–1142.
3. Section 2.8 presents the principal component analysis.
4. The structure of the paper needs to follow the journal's guidelines, e.g., Section 3 (which is missing in this version) should include the Conclusions. In addition, different spacing have been used in different sections.
5. Line 419. Explain why these concentrations were chosen.
6. Figure 1. It would be useful to add the photos with the other oils to illustrate the differences among them.
Comments on the Quality of English LanguageModerate editing of English language is required, e.g., "Table 5 shown the texture parameters ..." in line 351.
Author Response
14 December 2023
Dear Referee,
We would like to thank the referee for the close reading and for the proper suggestions. We hope that we provide all the answers to the reviewer’s comments.
Thank you very much for the recommendations to publish our paper entitled “Characterization of beeswax and brown rice wax oleogels based on different types of vegetable oils and their impact on wheat flour dough technological behavior during buns making”.
The present version of the paper has been revised according to the reviewer’s suggestions.
We uploaded the corrected version of the article for which we used the red color for the addition text and strikethrough for the delete text.
Reviewer comments
Reviewer: This paper describes the use of oleogels with beeswax and brown rice bran wax in a dough recipe for bakery products. The authors analyze the rheolegical and textural properties of the oleogels and dough. The paper is well-written and it deals with a very interesting field, which the use of oleogels in dough. I believe that it will be of interest to the readers of this journal. Having said that, there are some issues that need to be addressed in order to improve the quality and readability of the paper.
Response: We want to thank to the referee for the close reading of our manuscript. We revised the manuscript according to the referee suggestions. Also, would like to thank the reviewer for all the comments and suggestions which have helped us to improve our paper.
Reviewer: In lines 92 - 97, the authors state that there are only a few works about the effects of oleogels on the textural and rheological characteristics of yeast-leavened food systems such as breads. However, the authors need to state what's the novelty of their work compared to these works.
Response: We would like to thank to the referee for his/her suggestions. We improved the introduction part (including lines 92-97) with more aspects regarding the novelty of our work compared to others works.
Reviewer: Except for the references for oleogels in dough, the paper should include a paragraph about oleogels with waxes. Some of the latest works that can be used as a starting point are the following:
- Frolova, Y.; Sarkisyan, V.; Sobolev, R.; Makarenko, M.; Semin, M.; Kochetkova, A. The influence of edible oils’ composition on the properties of beeswax-based oleogels. Gels 2022, 8, 48.
- Dimakopoulou-Papazoglou, D.; Giannakaki, F.; Katsanidis, E. Structural and Physical Characteristics of Mixed-Component Oleogels: Natural Wax and Monoglyceride Interactions in Different Edible Oils. Gels 2023, 9, 627.
- Winkler-Moser, J.K.; Anderson, J.; Felker, F.C.; Hwang, H.-S. Physical properties of beeswax, sunflower wax, and candelilla wax mixtures and oleogels. J. Am. Oil Chem. Soc. 2019, 96, 1125–1142.
Response: We would like to thank the referee for his/her remarks. We revised the introduction part with paragraphs about oleogels with waxes. Also, we used the recommended bibliography to improve our introduction part.
Reviewer: Section 2.8 presents the principal component analysis.
Response: We would like to thank the referee for his/her remarks.
Reviewer: The structure of the paper needs to follow the journal's guidelines, e.g., Section 3 (which is missing in this version) should include the Conclusions. In addition, different spacing have been used in different sections.
Response: We want to thank the referee for his/her close reading of our manuscript. We moved the conclusion part to section 3, according to the journal's guidelines.
Reviewer: Line 419. Explain why these concentrations were chosen.
Response: We completed our manuscript with the explanations why these concentrations were chosen.
Reviewer: Figure 1. It would be useful to add the photos with the other oils to illustrate the differences among them.
Response: We want to thank the referee for his/her suggestions. We deleted figure 1 from our manuscript.
Reviewer: Comments on the Quality of English Language
Moderate editing of English language is required, e.g., "Table 5 shown the texture parameters ..." in line 351
Response: We revised the English language from our manuscript.
Sincerely,
Georgiana Codină et co.
Reviewer 3 Report
Comments and Suggestions for Authors
The authors should in-depth revise this manuscript. The authors claim to evaluate the replacement of shortening in a bun recipe but the basic reference standard does not contain a shortening? Moreover, the authors mainly describe results rahter than also interpreting the results and also build an mechanistic understanding of the studied bun model system. Graphic representation should be improved, now DSC graphs just come directly from the software program. Please work on them first before putting them in a scientific paper. The same holds for the rheology data.
Line 68: 10% sunflower oil? What is the other 90%, wax?
Figure 1: what is this showing: very non-informing figure, delete
Figure 2: you have a lot of variability in the results, why, this is not a normal frequency sweep curve
Line 128: similar results? First it is mentioned that the RW gel had a very low frequency dependence and the results of Patel showed a frequency dependent behavior?
Line 134: how is this calculated?
Line 174: what is the first cooling profile?
Line 175: crystallization of the olive oil?
Line 216: SEM?
Line 220: is there a microscopy technique used? where do you see larger aggregates?
Line 226: s the hardness of the oleogel as such relevant if you are looking at the use of the oleogels in a leavened dough? What is the link between the hardness of the oleogel and the hardness of the dough?
Line 329: Lipids will intrecact with starch, breaking it continuity struc-329 ture, reducing starch gelatinization and swelling, leading to a low dough consistency. -> this statement is not correct, better check the science behind your statements!
Line 338: Control is without any lipids, so this can also be the case when using the traditional shortening/margarine?
storage of samples: Storage temperature? Was the time between production and analysis always the same for the different blends?
what was your cooling rate of the samples?
Comments on the Quality of English Language
The document needs to be revised thoroughly for English language and flow of writing. It is not easily readable and lacks a coherent structure of representation and discussion.
Author Response
14 December 2023
Dear Referee,
We would like to thank the referee for the close reading and for the proper suggestions. We hope that we provide all the answers to the reviewer’s comments.
Thank you very much for the recommendations to publish our paper entitled “Characterization of beeswax and brown rice wax oleogels based on different types of vegetable oils and their impact on wheat flour dough technological behavior during buns making”.
The present version of the paper has been revised according to the reviewer’s suggestions.
We uploaded the corrected version of the article for which we used the red color for the addition text and strikethrough for the delete text.
Reviewer comments
Reviewer: The authors should in-depth revise this manuscript. The authors claim to evaluate the replacement of shortening in a bun recipe but the basic reference standard does not contain a shortening? Moreover, the authors mainly describe results rahter than also interpreting the results and also build an mechanistic understanding of the studied bun model system. Graphic representation should be improved, now DSC graphs just come directly from the software program. Please work on them first before putting them in a scientific paper. The same holds for the rheology data.
Response: We want to thank to the referee for the close reading of our manuscript. We revised the manuscript according to the referee suggestions. We added in the manuscript a refrence which contain shortening and we improved the graphs from our mansucript. Also, would like to thank the reviewer for all the comments and suggestions which have helped us to improve our paper.
Reviewer: Line 68: 10% sunflower oil? What is the other 90%, wax?
Response: We revised all the introduction section, and this part has been deleted from it.
Reviewer: Figure 1: what is this showing: very non-informing figure, delete
Response: We deleted the figure according to the referee’s suggestions.
Reviewer: Figure 2: you have a lot of variability in the results, why, this is not a normal frequency sweep curve
Response: We would like to thank the referee for his/her remarks. We changed the figure and we put the moduli variability only from 0.1 to 10 Hz. The frequency range 0.1-100Hz was chosen to demonstrate the behavior of oleogels outside the linearity range. The evolution of the storage (G′) and loss (G″) moduli in the frequency range 0.1-10 Hz had a linear behavior and when applying a shear stress above 10 Hz, it had a non-linear response. Similar articles that also describe the behavior of oleogels outside the linear domain are:
- Martín-Alfonso, M. A., Rubio-Valle, J. F., Hinestroza, J. P., & Martín-Alfonso, J. E. (2022). Impact of vegetable oil type on the rheological and tribological behavior of Montmorillonite-Based oleogels. Gels, 8(8), 504.
- Wijarnprecha, K., de Vries, A., Santiwattana, P., Sonwai, S., & Rousseau, D. (2019). Microstructure and rheology of oleogel-stabilized water-in-oil emulsions containing crystal-stabilized droplets as active fillers. Lwt, 115, 108058.
- Guo, K., Zhu, Y., Wang, J., Jiang, C., & Yu, J. (2019, April). Characterizing the viscoelastic properties of a tissue mimicking phantom for ultrasound elasticity imaging studies. In IOP Conference Series: Materials Science and Engineering (Vol. 490, No. 2, p. 022035). IOP Publishing.
- Carberry, D. M., Baker, M. A. B., Wang, G. M., Sevick, E. M., & Evans, D. J. (2007). An optical trap experiment to demonstrate fluctuation theorems in viscoelastic media. Journal of Optics A: Pure and Applied Optics, 9(8), S204.
- Thakur, D., Singh, A., Prabhakar, P. K., Meghwal, M., & Upadhyay, A. (2022). Optimization and characterization of soybean oil-carnauba wax oleogel. LWT, 157, 113108.
- Pușcaș, A., Tanislav, A. E., Mureșan, A. E., Fărcaș, A. C., & Mureșan, V. (2022). Walnut oil oleogels as milk fat replacing system for commercially available chocolate butter. Gels, 8(10), 613.
Reviewer: Line 128: similar results? First it is mentioned that the RW gel had a very low frequency dependence and the results of Patel showed a frequency dependent behavior?
Response: We would like to thank the referee for the close reading of our manuscript. Indeed, some confusion has been made. We revised and underlined more clearly the similarity of our results with Patel et. results.
Reviewer: Line 134: how is this calculated?
Response: The paragraph has been reworded because no calculation formula was used to highlight the behavers of the oleogels in the temperature range at which the analysis was performed.
Line: 179 Similar results were obtained by Tavernier et al. for sunflower oil and hemp seed oil oleogels with beeswax. It correlates the chemical structure of beeswax which has long chain wax esters with a fatty acid portion between C20 and C24 and a fatty alcohol portion between C24 and C28 leading to wax crystal formation at lower temperatures than other oleogelators.
Reviewer: Line 174: what is the first cooling profile?
Response: According to the Materials and Methods section, DSC analysis was conducted, as follows: a first heating from 20 °C to 90 °C, then a cooling of sample from 90 °C to -60 °C, a second heating from -60 °C to 100 °C, and finally cooling to 20 °C. Therefore, two sequential heating and cooling profiles were recorded; this is what “first cooling profile” refers to.
Reviewer: Line 175: crystallization of the olive oil?
Response: The phrase at lines 178-181 was modified and also the last phrase of this paragraph was improved to explain better why this phenomenon was recorded. Table 1 was also modified because the second phase transition data was missing.
Reviewer: Line 216: SEM?
Line 220: is there a microscopy technique used? where do you see larger aggregates?
Response: The SEM technique have been used by us in a previous article: Ropciuc, S.; Dranca, F.; Oroian, M.A.; Leahu, A.; Codină, G.G.; Prisacaru, A.E. Structuring of Cold Pressed Oils: Evaluation of the Physicochemical Characteristics and Microstructure of White Beeswax Oleogels. Gels 2023, 9, 216. https://doi.org/10.3390/gels9030216
However, it may be confusing to the readers to correlate our data with some previous results. So, we deleted the paragraphs related to SEM results.
Reviewer: Line 226: s the hardness of the oleogel as such relevant if you are looking at the use of the oleogels in a leavened dough? What is the link between the hardness of the oleogel and the hardness of the dough?
Response: In our opinion, the oleogel hardness may be an important parameter from the technological point of view (storage, manipulation, packaging filling). It may be an important parameter for manufacturers. Many studies have been analyzed oleogels hardness. Some examples:
- Gaudino, N.; Ghazani, S.M.; Clark, S.; Marangoni, A.G.; Acevedo, N.C. Development of lecithin and stearic acid based oleogels and oleogel emulsions for edible semisolid applications. Food Res. Int. 2019, 116, 79–89.
- Li, J.X.; Guo, R.H.; Bi, Y.L.; Zhang, H.; Xu, X.B. Comprehensive evaluation of saturated monoglycerides for the forming of oleogels. Lwt-Food Sci. Technol. 2021, 151, 112061.
- Flöter, E.; Wettlaufer, T.; Conty, V.; Scharfe, M. Oleogels—Their Applicability and Methods of Characterization. Molecules 2021, 26, 1673. https://doi.org/10.3390/molecules26061673
- Choi, K.-O., Hwang, H.-S., Jeong, S., Kim, S. and Lee, S. (2020), The thermal, rheological, and structural characterization of grapeseed oil oleogels structured with binary blends of oleogelator. Journal of Food Science, 85: 3432-3441. https://doi.org/10.1111/1750-3841.15442
- da Silva, T.L.T., Arellano, D.B. and Martini, S. (2018), Physical Properties of Candelilla Wax, Monoacylglycerols, and Fully Hydrogenated Oil Oleogels. J Am Oil Chem Soc, 95: 797-811. https://doi.org/10.1002/aocs.12096
Reviewer: Line 329: Lipids will interact with starch, breaking it continuity structure, reducing starch gelatinization and swelling, leading to a low dough consistency. - this statement is not correct, better check the science behind your statements!
Response: We want to thank the referee for the close reading of our manuscript. We modified the entire paragraph.
Reviewer: Line 338: Control is without any lipids, so this can also be the case when using the traditional shortening/margarine?
Response: We want to thank the referee for his/her suggestions. We agree with the referee point of view, and we completed our manuscript with data related to dough samples with shortening.
Reviewer: storage of samples: Storage temperature? Was the time between production and analysis always the same for the different blends?
Response: The following text has been introduced
Line 527: The mixture obtained was poured into tubes for solidification. The oleogels were stored under refrigerated conditions at 4 ℃ for five days until analysis.
Reviewer: what was your cooling rate of the samples?
Response: Thank you for your observation, we completed the manuscript with the cooling rate of the samples.
Sincerely,
Georgiana Codină et co.